# A Hidden Semantic Bottleneck in Conditional Embeddings of Diffusion Transformers

**Trung X. Pham, Kang Zhang, Ji Woo Hong, Chang D. Yoo**[*]
Korea Advanced Institute of Science and Technology (KAIST)
{trungpx, zhangkang, jiwoohong93, cd_yoo}@kaist.ac.kr

## Abstract

Diffusion Transformers have achieved state-of-the-art performance in class-conditional and multimodal generation, yet the structure of their learned conditional embeddings remains poorly understood. In this work, we present the first systematic study of these embeddings and uncover a notable redundancy: class-conditioned embeddings exhibit extreme angular similarity, exceeding 99% on ImageNet-1K, while continuous-condition tasks such as pose-guided image generation and video-to-audio generation reach over 99.9%. We further find that semantic information is concentrated in a small subset of dimensions, with head dimensions carrying the dominant signal and tail dimensions contributing minimally. By pruning low-magnitude dimensions–removing up to two-thirds of the embedding space–we show that generation quality and fidelity remain largely unaffected, and in some cases improve. These results reveal a semantic bottleneck in Transformer-based diffusion models, providing new insights into how semantics are encoded and suggesting opportunities for more efficient conditioning mechanisms.

## 1 Introduction

Transformer-based diffusion models have recently emerged as state-of-the-art architectures for generative modeling tasks across diverse domains, including class-conditional image synthesis DiT (Peebles & Xie, 2023), MDT (Gao et al., 2023), SiT (Ma et al., 2024), LightningDiT (Yao et al., 2025), Model-Guidance (MG) (Tang et al., 2025), REPA (Yu et al., 2025), pose-guided person image generation (Pham et al., 2024), and video-to-audio generation (Pham et al., 2025b). These models combine the expressive capacity of Transformer backbones with diffusion processes to generate high-fidelity, semantically consistent outputs. A key component of such models is the conditional embedding vector, often formed by summing class label and timestep embeddings and injected via adaptive layer normalization (AdaLN). *Yet despite their state-of-the-art performance and broad adoption, the role and internal structure of these learned conditional embeddings remain poorly understood.*

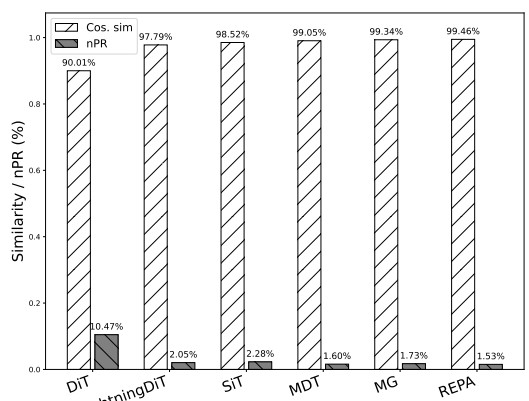

Figure 1: **Hidden Semantic Bottleneck: Extreme Alignment and Dimensional Sparsity.** Conditional vectors $\vec{c}$ in state-of-the-art Transformer diffusion models on ImageNet-1K exhibit very high pairwise cosine similarity (mostly 90–99%) while concentrating semantic information in only a few of 1,152 dimensions.

In this work, we present a systematic analysis of conditional embeddings in diffusion transformers and uncover two key findings. **(1)** Class-condition vectors exhibit extreme alignment, with cosine similarity exceeding 99% on ImageNet-1K across multiple state-of-the-art meth-

---

[*]Corresponding Author

Figure 2: Transformer-based diffusion models inject conditions as a globally compact vectors $\vec{v}_i$ via AdaLN for outputs such as images or mel-spectrograms.

ods (Fig. 1, white bar). **(2)** The learned conditional vector $\vec{c}$ is markedly sparse: only about 10–20 of its 1,152 dimensions carry substantial magnitude, yielding a normalized participation rate (nPR) of just 1–2% (Fig. 1, gray bar). When we prune up to 66% of dimensions and perform inference with the resulting sparsified $\vec{c}$, generation quality remains essentially unchanged, exposing significant over-parameterization. These findings challenge common assumptions about semantic conditioning and indicate that diffusion transformers encode conditioning signals far more compactly than previously believed, offering a new design perspective for generative models. *Our contributions are as follows:*

- **Extreme similarity.** We present the first systematic analysis showing that, in discrete class-conditional tasks (e.g., ImageNet), transformer-based diffusion models learn class-conditioned embeddings with up to 99% pairwise cosine similarity, and in continuous-condition tasks (e.g., pose-guided image or video-conditioned audio generation), the similarity exceeds 99.9%.

- **Sparse representations.** We find that semantic information is concentrated in a small set of embedding dimensions, while most remain near zero, revealing highly sparse conditional representations.

- **Redundancy and pruning.** We demonstrate that aggressively pruning low-magnitude dimensions preserves or even improves generation quality, highlighting substantial redundancy and enabling more efficient conditioning.

- **Mechanistic insight.** We provide hypotheses, supported by analyses and theoretical reasoning, to explain the emergence of high similarity, sparsity, and pruning effectiveness.

## 2 RELATED WORK

**Diffusion transformers and conditioning via AdaLN.** Diffusion models have progressed from U-Net backbones (Rombach et al., 2022) to transformer-based designs such as DiT (Peebles & Xie, 2023), SiT (Ma et al., 2024), LightningDiT (Yao et al., 2025), MG (Tang et al., 2025), X-MDPT (Pham et al., 2024), MDSGen (Pham et al., 2025b), and UCGM (Sun et al., 2025), achieving strong results across image, audio, and multimodal generation.

These models embed conditional signals–class labels, poses, or video features–into timestep embeddings and inject them via adaptive layer normalization (AdaLN) (Fig. 2), where condition vectors modulate all layers through learned scale–shift parameters. Unlike the distributed conditioning of U-Nets, this global AdaLN mechanism motivates our study of how semantic information is encoded in transformer conditional vectors.

**Prior work on conditional embedding analysis.** Li et al. (2023) examined activation sparsity in Transformers for NLP and ImageNet with classification, but systematic studies of conditional embeddings in generative diffusion remain scarce. Early efforts targeted U-Net conditioning (Rombach et al., 2022; Saharia et al., 2022), while transformer-based models focused on architectural or training advances (Peebles & Xie, 2023; Ma et al., 2024; Yu et al., 2025; Tang et al., 2025; Gao et al., 2023; Pham et al., 2024; 2025b). We fill this gap with an analysis of transformer conditional embeddings and their link to representation collapse.

**Collapse in contrastive learning.** Representation collapse–mapping diverse inputs to nearly identical embeddings–is well known in contrastive learning (Grill et al., 2020; Zbontar et al., 2021). We observe a related effect in diffusion transformers: conditional embeddings across classes reach extreme angular similarity (>99% cosine) without harming generation quality, indicating a distinct embedding usage compared to contrastive methods.

**Hyperspherical embeddings and compressed codes.** Our findings align with hyperspherical embedding (Liu et al., 2017) and information bottleneck theory (Tishby et al., 2000),

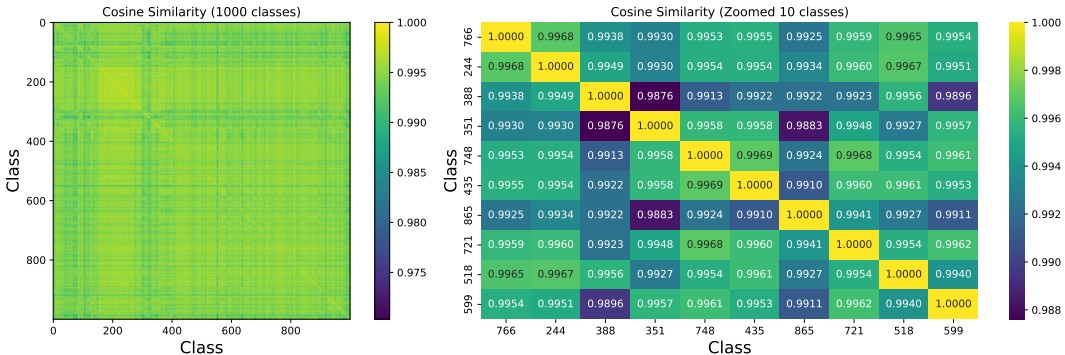

Figure 3: Cosine similarity of conditional vectors $\vec{c} = y + t$ across 1000 ImageNet classes using REPA-XL (Yu et al., 2025). Despite distinct semantics, embeddings show over 99% similarity for nearly all class pairs. **Left**: full $1000 \times 1000$ matrix showing global alignment. **Right**: zoomed $10 \times 10$ subset for randomly chosen classes. Additional results for other SOTA methods appear in the Appendix.

which describe semantic compression into low-dimensional subspaces. Similar trade-offs appear in VAEs and multimodal systems (Kingma & Welling, 2013; Tsai et al., 2019). Diffusion transformers further compress conditioning into a small set of active head dimensions, leaving others largely redundant.

**Conditioning injection: U-Net vs. Transformers.** U-Net diffusion models inject conditions at multiple spatial scales via concatenation or cross-attention (Rombach et al., 2022; Dhariwal & Nichol, 2021), allowing localized feature extraction. Transformers, in contrast, apply global AdaLN modulation, which likely drives the observed sparsity and high similarity in conditional embeddings as semantics collapse into a few dominant dimensions.

## 3 Emergent Property I: Near-Uniform Cosine Similarity

### 3.1 Setup

We systematically analyze six state-of-the-art diffusion transformer models–DiT (Peebles & Xie, 2023), MDT (Gao et al., 2023), SiT (Ma et al., 2024), REPA (Yu et al., 2025), LightningDiT (Yao et al., 2025), and Model-Guided (Tang et al., 2025)–using their official pretrained checkpoints released on GitHub (XL models). The primary analysis is conducted on ImageNet-1K, where we compute pairwise cosine similarity matrices across all class-conditioned vectors $\vec{c} \in \mathbb{R}^{1152}$. Each $\vec{c}$ is formed by summing the learned class embedding and timestep embedding, resulting in the final conditional vector injected into the denoising transformer backbone. To assess generality across domains, we extend the analysis to pose-guided image synthesis using X-MDPT (Pham et al., 2024) ($\vec{c} \in \mathbb{R}^{1024}$) and video-to-audio generation with MDSGen (Pham et al., 2025b) ($\vec{c} \in \mathbb{R}^{768}$), again utilizing publicly available pretrained weights. This consistent evaluation setup ensures reproducibility and enables direct comparison across models and tasks.

### 3.2 Cosine Similarity Heatmaps

Fig. 3 (left) shows the full 1,000-class cosine-similarity matrix, where trained models reach up to $\sim 99\%$ similarity across class pairs. For clarity, Fig. 3 (right) provides a zoomed-in view of 10 randomly sampled classes, revealing the same strong alignment. Additional results for other methods are provided in the Appendix.

### 3.3 Cross-Task Examination

The strong alignment of conditional embeddings extends beyond class-conditional image generation to pose-guided image synthesis and video-to-audio generation (Fig. 4). X-MDPT (Pham et al., 2024) and MDSGen (Pham et al., 2025b) exhibit extreme cosine similarity—up to 99.98% on DeepFashion and 99.99% on VGGSound—even with randomly varying test samples and conditions (e.g., persons, poses, videos). Because MDSGen shows patterns nearly identical to X-MDPT, its cosine heat map is omitted. This striking consistency indicates that diverse inputs yield almost identical embeddings before denoising; we examine possible explanations in the following sections.

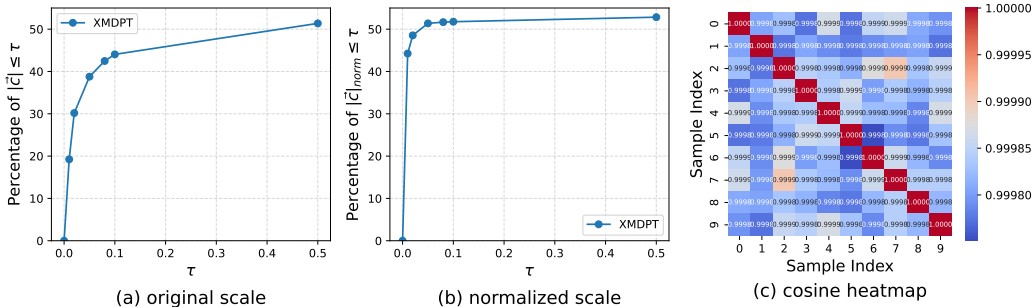

(a) original scale          (b) normalized scale         (c) cosine heatmap

Figure 4: **Sparsity and alignment of conditional embeddings in X-MDPT (Pham et al., 2024).** (a) and (b): With $\tau = 0.1$, over $51\%$ of components in the conditional vectors have magnitudes below the threshold, highlighting significant sparsity. Remarkably, pruning these dimensions has minimal effect on generation quality. (c) Cosine similarity between random test samples in DeepFashion exceeds $99.9\%$, confirming extreme alignment across conditional embeddings.

## 4 Emergent Property II: Sparse Magnitude Distribution

### 4.1 Magnitude Histograms

Fig. 5 shows the histogram of absolute component values of $\vec{c}$: only about 10–20 of the 1,152 dimensions exceed 0.1 in magnitude, and roughly 10 exceed 1. Fig. 6 visualizes the learned conditional vector for each method, underscoring its pronounced sparsity. For completeness, we include continuous tasks such as X–MDPT and MDSGen in the Appendix; their embeddings appear less sparse, consistent with the higher participation ratio (PR) reported in Tab. 1.

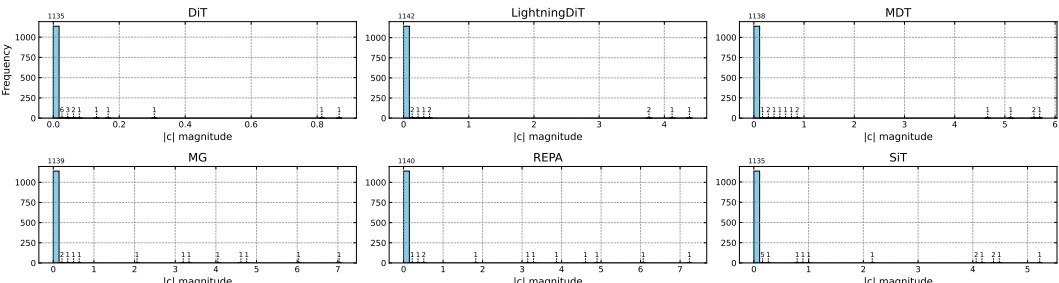

Figure 5: *Magnitude histogram distribution* of learned conditional vector embedding $\vec{c} \in \mathbb{R}^{\times 1152}$. Most dimensions have near-zero values ($< 0.01$), with only $\sim 5 - 20$ dimensions showing dominant magnitudes. This sparsity holds across multiple models, including DiT, MDT, LightningDiT, MG, SiT, and REPA. It is best viewed with $300\%$ zoomed in.

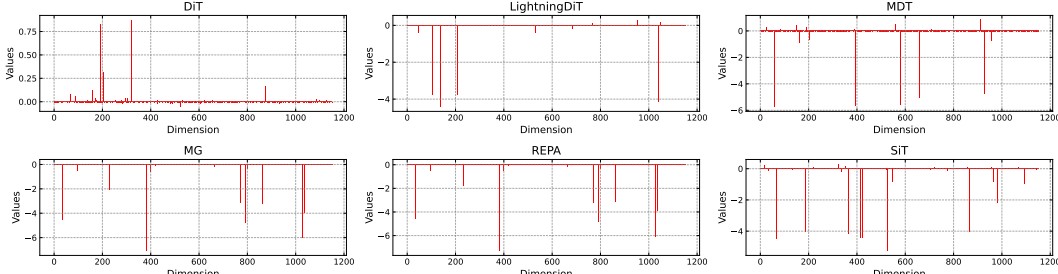

Figure 6: *Original distribution* of learned conditional vector embedding $\vec{c} \in \mathbb{R}^{1 \times 1152}$. Most dimensions have near-zero values ($< 0.01$), with only $\sim 5 - 20$ dimensions showing dominant magnitudes. This sparsity holds across multiple models, including DiT, MDT, LightningDiT, MG, SiT, and REPA. It is best viewed with $200\%$ zoomed in.

Table 1: Participation Ratio (PR) in learned conditional embeddings of state-of-the-art models on Imagenet-1K generation (discrete) and DeepFashion/VGGSound (continuous).

| Embedding Metrics | DiT | SiT | MDT | LightningDiT | MG | REPA | X-MDPT | MDSGen |
|---|---|---|---|---|---|---|---|---|
| Condition Dim ($d$) | 1152 | 1152 | 1152 | 1152 | 1152 | 1152 | 1024 | 768 |
| PR ($\alpha$) | 120.69 | **26.25** | **18.45** | **23.70** | **19.98** | **17.67** | 495.75 | 104.22 |
| nPR ($\alpha_{\text{norm}}$) | 10.47% | **2.28%** | **1.60%** | **2.05%** | **1.73%** | **1.53%** | 48.42% | 13.57% |
| Cosine Sim. ($cs$) | 0.9001 | 0.9852 | 0.9905 | 0.9779 | 0.9934 | 0.9946 | **0.9998** | **0.9999** |

## 4.2 DIMENSION CONTRIBUTION: PARTICIPATION RATIO

To quantify how many dimensions effectively contribute, we compute the **participation ratio** (PR) on absolute magnitudes $v_i = |c_i|$:

$$\alpha = \text{PR}(v) = \frac{\left(\sum_{i=1}^{d} v_i\right)^2}{\sum_{i=1}^{d} v_i^2}, \quad \alpha_{\text{norm}} = \frac{\alpha}{d} \text{ , with d is dimension.} \tag{1}$$

PR estimates the number of coordinates carrying most of the total magnitude. Tab. 1 shows that state-of-the-art models (MDT, LightningDiT, MG, REPA) rely on less than 2% of dimensions, whereas continuous tasks such as X-MDPT and MDSGen use a larger fraction (13–48%) and exhibit even higher cosine similarity (up to 99.99% vs. 90–99.4%).

This suggests that continuous-condition embeddings both engage more dimensions and distribute information more uniformly, naturally leading to stronger alignment across samples compared to discrete class-conditional ImageNet generation.

## 5 FROM OBSERVATION TO ACTION: PRUNING REDUNDANT DIMENSIONS

**Role of tail dimensions.** To quantify sparsity and effective dimensionality, we define the **sparsity ratio** at threshold $\tau$ as

$$s_{\text{tail}(\tau)} = \frac{1}{d}\#\{i : |c_i| < \tau\}. \tag{2}$$

With $\tau = 0.01$, we observe a sparsity ratio of $s \approx 0.38$–$0.40$. We define the high-magnitude "head" as $s_{\text{head}}(\tau) = \frac{1}{d}\#\{i : |c_i| > \tau\}$ and the low-magnitude "tail" as the remaining coordinates. Using REPA as a representative model, we progressively prune $\vec{c}$ at thresholds $\tau \in \{0.01, 0.02, \dots\}$ and find

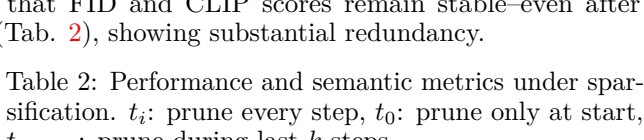

Figure 7: **Class-conditioned image generation with head removal.** ImageNet samples after pruning top-magnitude dimensions of $\vec{c}$ (threshold $\tau$); removing only a few head dimensions markedly degrades quality.

that FID and CLIP scores remain stable–even after removing up to 66% of dimensions (Tab. 2), showing substantial redundancy.

Pruning at $\tau = 0.01$ (removing ∼38% of dimensions) preserves or improves image quality (Fig. 8, Tab. 2). Pruning during late denoising steps yields larger FID gains and modest CLIP improvements, aligning with the gradual rise in cosine similarity toward the final steps. For consistency, we report statistics using the conditional vector at the initial step $t_0$ and discuss possible explanations in the

Table 2: Performance and semantic metrics under sparsification. $t_i$: prune every step, $t_0$: prune only at start, $t_{n-k,n}$: prune during last $k$ steps.

| | Threshold $\tau$ | # Removed Dims | FID ↓ | IS ↑ | CLIP↑ |
|---|---|---|---|---|---|
| Prune | Baseline (REPA) | 0/1152 (0%) | 7.1694 | **176.02** | 29.746 |
| Tail | $\tau = 0.01$ ($t_i$) | 448/1152 (38.94%) | 7.2143 | 171.99 | 29.737 |
| | $\tau = 0.01$ ($t_0$) | 448/1152 (38.94%) | **7.1690** | 175.97 | **29.807** |
| | $\tau = 0.01$ ($t_{n-k,n}$) | 448/1152 (38.94%) | **7.1598** | 175.49 | **29.805** |
| | $\tau = 0.02$ ($t_i$) | 762/1152 (66.21%) | 9.2202 | 125.15 | 29.221 |
| | $\tau = 0.05$ ($t_i$) | 1110/1152 (96.41%) | 56.2308 | 20.47 | 22.177 |
| | $\tau = 5.0$ ($t_i$) | 1149/1152 (99.80%) | 356.135 | 1.77 | 21.922 |
| Head | $\tau = 5.0$ ($t_i$) | 2/1152 (0.20%) | 7.8478 | 164.15 | 29.555 |
| | $\tau = 1.0$ ($t_i$) | 8/1152 (0.69%) | 523.7637 | 1.95 | 22.690 |

following sections. Next, we examine in detail how head dimensions influence generation quality and clarify how their role differs from that of the tail dimensions.

**Role of head dimensions.** As shown in Fig. 7, removing only a few high-magnitude dimensions (e.g., 4-6/1152) dramatically degrades generation quality. In contrast, pruning up

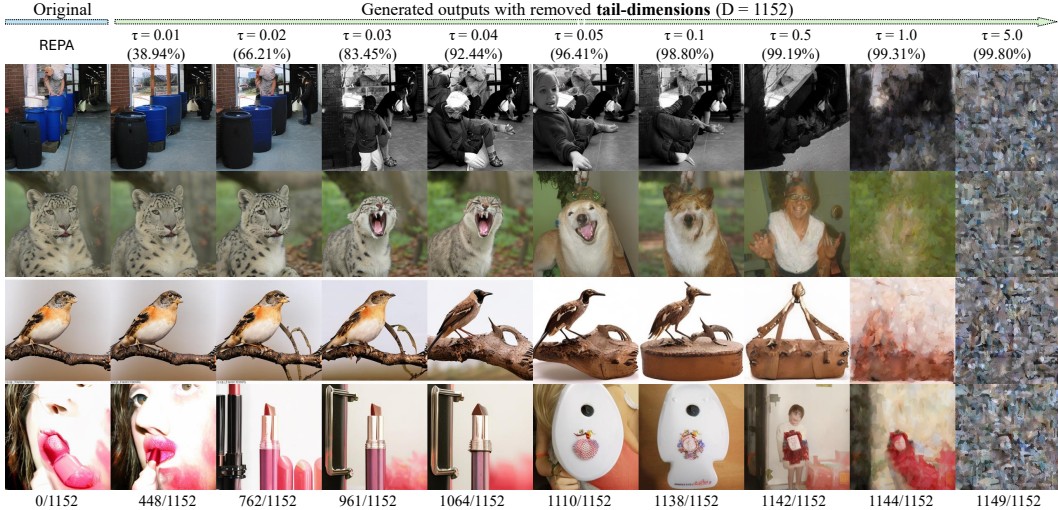

Figure 8: **Class-conditioned image generation (tail removal).** ImageNet samples with progressive removal of low-magnitude dimensions in $\vec{c}$ (threshold $\tau$ on absolute value). Image quality remains high or better baseline REPA even when 38–>80% of dimensions are pruned (generated images in the second column), as long as key head dimensions are retained.

to 66% of low-magnitude tail dimensions (762/1152) leaves quality largely intact. Variance analysis (Fig. 9) further reveals that only ∼15–20 head dimensions carry most of the variance, particularly across classes, highlighting their critical role compared to the tails.

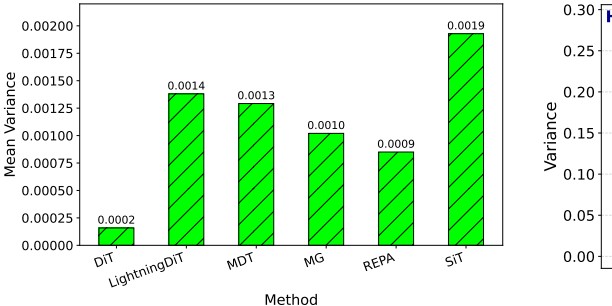

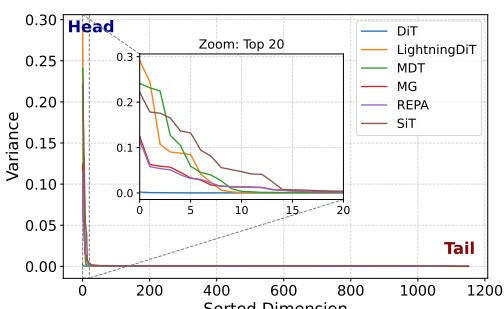

(a) Average variance across dimensions.

(b) Variance per-dimension (sorted).

Figure 9: **Variance concentration in conditional vectors $\vec{c}$.** (a) Mean variance across models stays non-zero, showing no collapse. (b) Variance is concentrated in only 15–20 head dimensions (<2%), while the remaining 98% of tail dimensions show minimal variation, indicating that semantic information is confined to a small subspace.

**Continuous task.** We apply the same pruning procedure to pose-guided person image generation with X-MDPT (Pham et al., 2024) and observe consistent behavior (Fig. 10). Unlike class-conditional ImageNet, this task requires a slightly higher threshold to induce similar sparsity: $\tau = 0.1$ yields $s \approx 0.38$–0.40 (Fig. 4 a,b) while preserving generation quality (Fig. 10). More qualitative results are available in the Appendix.

# 6 Underlying Mechanisms Behind Similarity, Sparsity, and Pruning: Hypotheses

## 6.1 How Can a Model Generate Correct Outputs Despite High Similarity?

Although conditional vectors exhibit high pairwise cosine similarity, our variance analysis (Fig. 9a) shows no embedding collapse: the mean variance across models remains small but clearly non-zero (0.0002–0.0019). This contrasts with the feature collapse often seen in contrastive learning, where embeddings converge to a single point and variance approaches

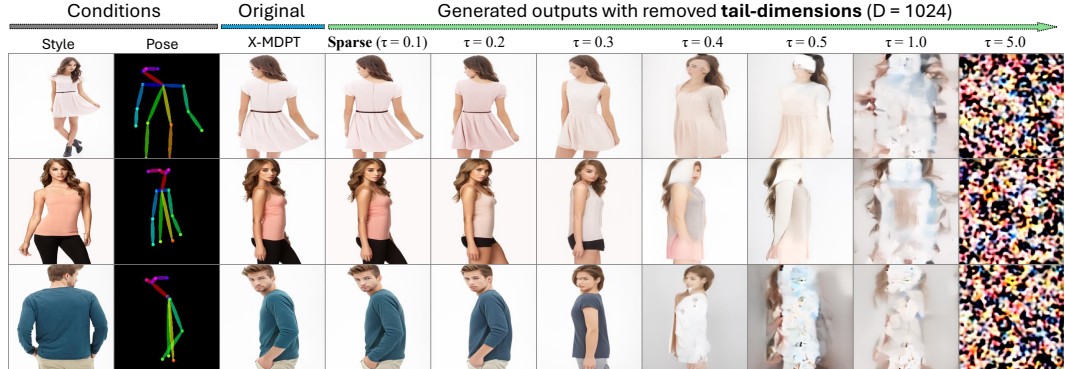

Figure 10: **Pose-conditioned person image generation.** *DeepFashion samples under different sparsification of $\vec{c}$.* $\tau$ is the magnitude threshold for pruning. Person-consistent images remain high quality even when 50–75% of dimensions are zeroed, as long as key head dimensions are preserved. Best viewed at 200% zoom; more samples appear in the Appendix.

zero (Wang & Isola, 2020; Zhang et al., 2022). Further, the dimension–threshold analysis in Fig. 11 reveals that with $\tau \approx 0.01$, a large fraction (50–90%) of dimensions retain low magnitude, saturating near $\tau = 0.02$, indicating that informative components are broadly distributed rather than concentrated in a few directions.

We hypothesize that diffusion transformers preserve this subtle but global structure because each denoising step predicts fine-grained Gaussian noise, providing a rich and stable training signal. Consequently, even when conditional embeddings lie within a narrow cone in feature space, their nuanced directional differences–amplified through adaptive layer normalization, the expressive Transformer backbone, and iterative refinement–remain sufficient to guide accurate class-conditional, pose-guided, and video-conditioned generation. *A deeper theoretical explanation of this robustness remains an open problem, calling for rigorous analysis in future work.*

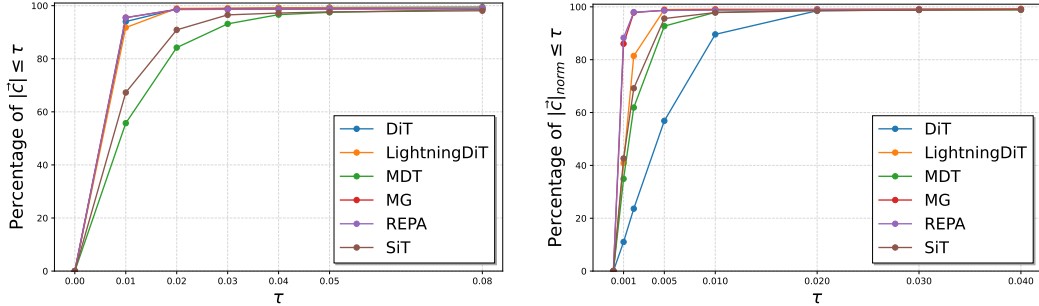

(a) Dimensions below threshold $\tau$ (original scale).   (b) Dimensions below threshold $\tau$ (normalized).

Figure 11: **Sparsity of conditional embeddings.** With $\tau = 0.01$, over 80% of components in the mean conditional vector fall below the threshold, revealing a highly sparse representation that starts to saturate near $\tau = 0.02$.

## 6.2 HYPOTHESIZING THE STRUCTURE OF CONDITIONAL EMBEDDINGS

**High Cosine Similarity.** We hypothesize that the extreme cosine similarity among class embeddings arises from the dynamics of diffusion Transformer training (Fig. 12 top). Since the model conditions on embeddings across all timesteps $t$, it favors embeddings that provide a stable, robust signal for denoising, resulting in globally aligned embeddings:

$$\text{cosine}(c_y, c_{y'}) \approx 0.99 \quad \forall y \neq y'. \tag{3}$$

Despite this high similarity, semantic differences are encoded in a small subset of high-magnitude *head* dimensions,

$$c_y = c_{y,\text{head}} + c_{y,\text{tail}}, \quad \|c_{y,\text{head}}\| \gg \|c_{y,\text{tail}}\|, \tag{4}$$

are sufficient to modulate Adaptive LayerNorm parameters $\gamma(c_y), \beta(c_y)$:

$$\gamma(c_y) = W_\gamma c_y, \quad \beta(c_y) = W_\beta c_y. \tag{5}$$

These subtle differences are progressively amplified by the iterative diffusion process, enabling correct and high-quality class-conditional generation despite the high overall cosine similarity.

**Observed sparsity in learned embeddings.** Conditional embeddings are highly sparse: for $d = 1152$, only about 1–2% of dimensions reach large magnitudes ($\approx$ 5–8), while most remain near zero ($10^{-3}$–$10^{-1}$). This head–tail pattern indicates that semantic information resides in a small subspace, aligning with our pruning results. We quantify sparsity using the normalized participation ratio $\alpha_{\text{norm}}$, which confirms that the effective dimensionality is far below $d$. As shown in Fig. 12 (bottom), monitoring this metric (nPR) while training the REPA B-2 model on ImageNet-1K for 200k steps shows a drop from about 90% early in training to under 6%, with the decline continuing–*revealing a natural sparsification dynamic in diffusion transformers*. Extended analyses appears in the Appendix.

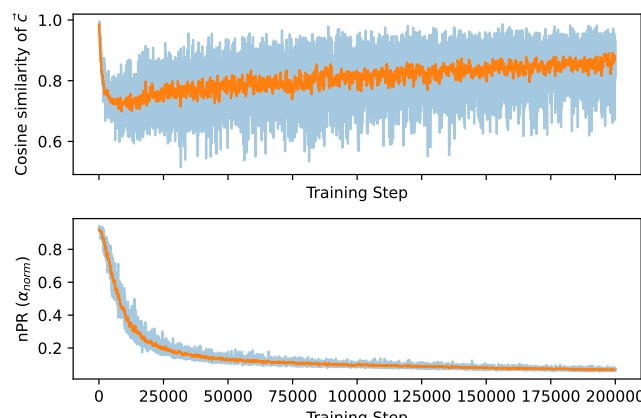

Figure 12: **Training dynamics of conditional embeddings.** Top: batchwise cosine similarity of $\vec{c}$ during training. Bottom: participation ratio (nPR) over training steps, showing progressive sparsification.

**Pruning Improves Generation.** We conjecture that pruning low-magnitude embedding dimensions acts as a form of noise suppression in diffusion Transformers. Let the conditional embedding be $c \in \mathbb{R}^d$ and decompose it as

$$c = c_{\text{head}} + c_{\text{tail}}, \quad \|c_{\text{head}}\| \gg \|c_{\text{tail}}\|. \tag{6}$$

In practice $c$ is mapped to Adaptive LayerNorm parameters $\gamma(c), \beta(c)$ that modulate hidden states $h$:

$$\text{AdaLN}(h \mid c) \; = \; \gamma(c) \odot \frac{h - \mu(h)}{\sigma(h)} + \beta(c). \tag{7}$$

Because semantic information concentrates in $c_{\text{head}}$, the tail $c_{\text{tail}}$ contributes only weak, low-variance signals (as shown before in Fig. 9b). Supporting this view, Fig. 13 visualizes class embeddings: when only head dimensions are retained, class clusters remain well separated, whereas tail-only embeddings collapse into an entangled cloud. Retaining these noisy tail dimensions can perturb $\gamma(c), \beta(c)$ and inject interference into the denoising trajectory, particularly in later inference steps where precision is critical.

We empirically observe that pruning (zeroing out) $c_{\text{tail}}$ at the initial step $t_0$ or at the final steps preserves generation quality, with late-step pruning yielding the strongest FID improvements. This supports the view that pruning suppresses interference and sharpens the semantic subspace. A more detailed analysis is provided in the Appendix.

## 7 DISCUSSION

Our results reveal a semantic bottleneck in transformer-based diffusion models: conditional embeddings place most semantic content in a small set of high-magnitude dimensions, leaving the majority near zero and largely redundant. For class-conditional ImageNet generation, this effect is strongest, with only a few dominant dimensions and a very low normalized participation ratio (nPR). Continuous-condition tasks (e.g., pose-guided image or video-to-audio generation) show a milder form of this sparsity, exhibiting more high-magnitude

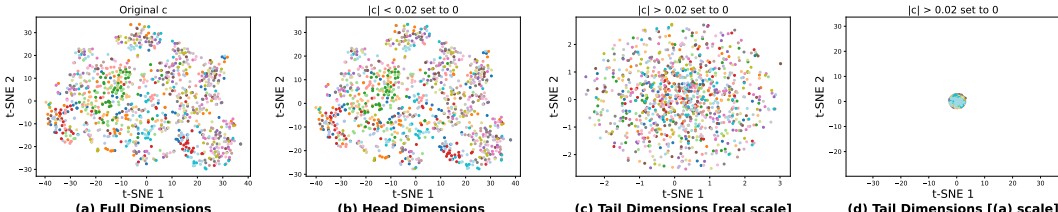

Figure 13: **t-SNE of class embeddings by head vs. tail dimensions.** Keeping only head dimensions (b) preserves clear class clusters similar to the full embedding (a), while tail-only embeddings (c,d) collapse into entangled points, revealing weak semantic structure. Results are from SiT-XL on ImageNet-1K; similar trends appear in other models.

dimensions and higher nPR values. The consistency of this pattern across architectures and tasks points to a general property of how diffusion transformers encode conditioning signals.

**Relation to Contrastive Learning Collapse** The observed alignment of conditional embeddings bears resemblance to representation collapse in contrastive learning methods like SimCLR Chen et al. (2020), SimSiam (Chen & He, 2021), BYOL (Grill et al., 2020), VICReg (Bardes et al., 2022), and Barlow Twins (Zbontar et al., 2021). In those settings, collapse leads to trivial embeddings and degraded downstream performance unless variance-promoting regularizers or repulse components (negative samples) are used (Zhang et al., 2022). Interestingly, diffusion transformers avoid such pitfalls: despite extreme angular similarity, they maintain strong generation quality. We hypothesize that AdaLN amplifies high-magnitude dimensions sufficiently to preserve semantic distinctiveness during denoising, and that diffusion models' iterative refinement mitigates the impact of collapsed embeddings.

**Why high cosine similarity occurs only in transformers**. Based on additional experiments with U-Net models, we clarify that high cosine similarity arises primarily in transformers, and not in U-Nets when timestep embeddings are removed. However, similar redundancy emerges in U-Net diffusion models once timestep embeddings are included. This distinction appears tied to conditioning mechanisms: AdaLN in transformers promotes compression into dominant dimensions, whereas U-Nets use concatenation or cross-attention, preserving richer representations.

**Relation to information bottleneck and AdaLN**. The sparsity mirrors information bottleneck behavior (Tishby et al., 2000), where networks distill essential features. AdaLN's linear scaling amplifies a few dominant dimensions, rendering others redundant. **Implications and risks**. Compact embeddings may conflate unrelated semantics (cosine similarity $\neq$ semantic similarity). This could limit controllability in multi-conditional tasks or fine-grained editing.

**Broader impact**. Our observations of extreme similarity, sparse embeddings, and effective pruning in transformer-based diffusion models suggest that similar redundancy patterns could also appear in other generative frameworks, such as U-Net diffusion models (when timestep embeddings are included), GANs, or autoregressive models. This points to a potential principle of compact and efficient conditioning, which may inspire future work on lighter models and interpretable embeddings across tasks and modalities.

## 8 Conclusions

We have uncovered an interesting phenomenon in transformer-based diffusion models: extreme angular similarity and semantic sparsity in conditional embeddings. Our extensive analyses reveal that only a small subset of high-magnitude dimensions carry semantic information, while the majority of dimensions are redundant. Despite this, diffusion transformers maintain robust generation quality even when up to 66% of the conditional vector is pruned or masked. These findings suggest a fundamental overparameterization of conditional encoding and motivate rethinking conditioning mechanisms for efficiency and interpretability. Future architectures could benefit from compressed or hybrid conditioning strategies that maintain semantic fidelity while reducing computational overhead. Exploring these directions may lead to more controllable, efficient, and versatile generative models across vision, audio, and multimodal domains.

## Acknowledgment

This work was supported by the Institute for Information & Communications Technology Planning & Evaluation (IITP) grant funded by the Korea government (MSIT) (No. RS-2021-II211381, Development of Causal AI through Video Understanding and Reinforcement Learning, and Its Applications to Real Environments) and (No. RS-2022-II220184, Development and Study of AI Technologies to Inexpensively Conform to Evolving Policy on Ethics) and partially supported by the KAIST Jang Young Sil Fellow Program.

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

# A   APPENDIX

In this appendix, we provide the details of experimental setups as well as more comprehensive analysis results for various approaches.

## A.1   SETUP DETAILS

We generate 5,000 samples from the public checkpoints of each method (5 samples per ImageNet class) and evaluate FID and IS using the LightningDiT (Yao et al., 2025) evaluation code. Inference follows the default hyperparameters and sampling steps specified by each model, using XL-size variants when available. For continuous tasks (X-MDPT and MDSGen with AdaLN), we adopt their largest released models (L-size and B-size, respectively). During inference, we modify only the conditional vector $\vec{c}$, keeping all other components unchanged.

## A.2   EXPLANATION HYPOTHESIS (EXTENDED)

**Pruning Improves Generation.**   We retain the decomposition used throughout the paper:
$$c = c_{\text{head}} + c_{\text{tail}}, \quad \|c_{\text{head}}\| \gg \|c_{\text{tail}}\|.$$
Here $c_{\text{head}}$ denotes the high-variance, semantically informative dimensions, while $c_{\text{tail}}$ corresponds to low-magnitude, low-variance dimensions.

Conditioning is implemented through Adaptive Layer Normalization (AdaLN). For a hidden activation $h \in \mathbb{R}^d$:
$$\text{AdaLN}(h \mid c) = \gamma(c) \odot \frac{h - \mu(h)}{\sigma(h)} + \beta(c),$$

with linear projections
$$\gamma(c) = W_\gamma c, \qquad \beta(c) = W_\beta c.$$

Linearity implies that
$$\gamma(c) = \gamma(c_{\text{head}}) + \gamma(c_{\text{tail}}), \qquad \beta(c) = \beta(c_{\text{head}}) + \beta(c_{\text{tail}}).$$
Empirically, $\text{Var}[\gamma(c_{\text{tail}})]$ and $\text{Var}[\beta(c_{\text{tail}})]$ are negligible compared to their head counterparts. However, we hypothesize that these weak terms can propagate as noise through the denoising trajectory, with a potentially larger effect in later inference steps ($t \to 0$), where precision is critical.

Define a pruning operator $\mathcal{P}(\cdot)$ that zeros out tail dimensions:
$$c' = \mathcal{P}(c) = c_{\text{head}}.$$

Pruning can be applied either at the initial step $t_0$ or during the final few steps of inference. While early-step pruning reduces redundancy early, we empirically observe that late-step pruning consistently yields stronger improvements in FID, supporting the hypothesis that late-step pruning suppresses residual noise and sharpens semantic guidance:
$$\text{AdaLN}(h \mid c') = \gamma(c_{\text{head}}) \odot \frac{h - \mu(h)}{\sigma(h)} + \beta(c_{\text{head}}).$$

Thus, pruning acts as an effective noise filter: removing weak tail dimensions reduces interference while focusing conditioning on dominant semantic directions, explaining why pruning preserves or can even improve generative quality.

**High Cosine Similarity.**   We empirically observe that the cosine similarity between class embeddings remains extremely high ($> 0.99$) across nearly all timesteps of denoising. We hypothesize that this is a consequence of dynamic training in diffusion Transformers: conditioning is applied across all timesteps $t$, and the network learns to maintain a stable, robust signal. This encourages embeddings to align along similar directions, while semantic distinctions are preserved in a small subspace of head dimensions:
$$c_y = c_{y,\text{head}} + c_{y,\text{tail}}, \quad \|c_{y,\text{head}}\| \gg \|c_{y,\text{tail}}\|.$$

Even with globally aligned embeddings, the high-magnitude head dimensions provide sufficient directional cues to modulate Adaptive LayerNorm parameters:

$$\gamma(c_y) = W_\gamma c_y, \quad \beta(c_y) = W_\beta c_y.$$

These small differences are progressively amplified by the iterative denoising process.

Thus, while embeddings appear nearly parallel in the full space, the effective semantic subspace defined by the head dimensions ensures that generation remains accurate and high-quality. This also explains why pruning tail dimensions, which contain low-magnitude, redundant signals, does not harm generation and can sometimes improve quality.

### A.3 Analysis of Embedding Sparsity

**Empirical observation.** In the pretrained diffusion Transformer embeddings we analyze ($d = 1152$), only a small subset of dimensions—about 1% to 2%—exhibit large absolute values (typical magnitude $\sim 5$–$8$), while the rest remain near-zero (typical magnitude $\sim 10^{-3}$–$10^{-1}$). We refer to the large-magnitude coordinates as the *head* and the rest as the *tail*.

**Metrics.** To quantify sparsity and effective dimensionality, we use the following statistics:

- **Sparsity ratio** at threshold $\tau$:

$$s(\tau) = \frac{1}{d} \#\{i : |c_i| > \tau\}.$$

  With $\tau$ set to a small constant (e.g., 0.5) this yields $s \approx 0.01$–$0.02$ empirically.

- **Participation ratio** (PR) on absolute magnitudes $v_i = |c_i|$ (a measure of effective dimensions):

$$\alpha = \mathrm{PR}(v) = \frac{\left(\sum_{i=1}^{d} v_i\right)^2}{\sum_{i=1}^{d} v_i^2}, \quad \alpha_{\mathrm{normalized}} = \frac{1}{d} \times \mathrm{PR}(v).$$

  PR gives an estimate of how many coordinates carry most of the total magnitude; we find $\mathrm{PR} \ll d$ (order tens).

This normalization maps the range of values to $\alpha_{\mathrm{normalized}} \in (0, 1]$, nPR ($\alpha_{\mathrm{normalized}}$) = 1 when all $d$ coordinates contribute equally. nPR $\approx k/d$ when effectively only $k$ coordinates carry the magnitude.

**Interpretation hypotheses.** We offer several plausible, non-exclusive explanations for this sparse phenomenon:

1. **Projection and scale effects.** The learned linear projections $W_\gamma, W_\beta$ (and any subsequent layers) can amplify a few coordinates of $c$ if their corresponding projection weights are large, producing a few dominant coordinates in the final modulation parameters.

2. **Stable conditioning across timesteps.** Because conditioning is applied across many timesteps, the optimizer favors a stable, low-dimensional conditioning signal to avoid disturbing denoising dynamics; encoding semantics in a few robust axes avoids noisy, volatile conditioning.

3. **Implicit sparsity from optimization/regularization.** Weight decay, initialization, and training dynamics may implicitly encourage small-magnitude coordinates; only the coordinates providing robust semantic signal are driven to large magnitudes.

**Consequences for AdaLN modulation.** Since AdaLN uses $\gamma(c) = W_\gamma c$ and $\beta(c) = W_\beta c$, large entries in $c$ dominate the modulation:

$$\gamma(c) \approx W_\gamma c_{\mathrm{head}}, \qquad \beta(c) \approx W_\beta c_{\mathrm{head}}.$$

Thus, the denoising network effectively receives conditioning from a low-dimensional subspace, explaining why zeroing most coordinates (sparsification) has a small empirical impact and why pruning can even improve performance by removing weak, noisy contributions.

## A.4 More Visualizations of Other Methods

### A.4.1 Cosine Similarity with Pairwise Analysis.

For completeness, we present full pairwise cosine similarity matrices for all six state-of-the-art diffusion transformer models evaluated on ImageNet. Each matrix reports the cosine similarity between conditional embeddings for every pair of ImageNet-1K classes, offering a comprehensive view of how uniformly aligned these vectors are across the label space. The results reinforce the main paper's findings: near-uniform similarity is pervasive across models and classes, with the sole exception of DiT, whose lowest pairwise similarity is about 88%. Notably, DiT also delivers weaker generative performance (higher FID) compared to the other models, further distinguishing it from the rest of the evaluated methods.

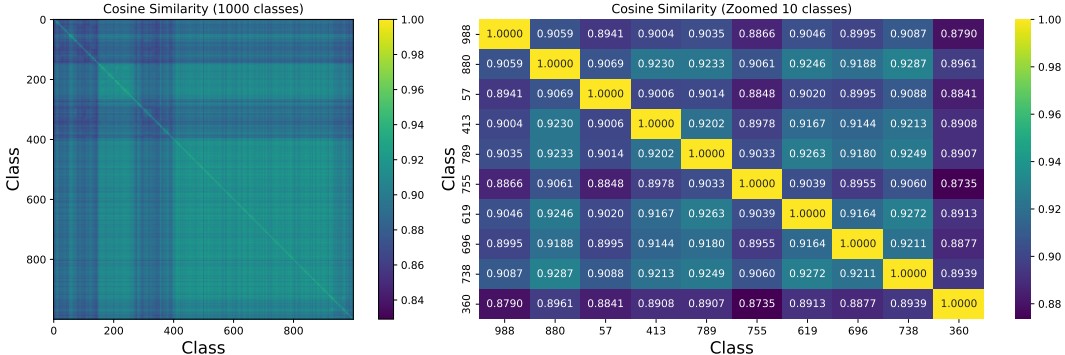

Figure 14: Cosine similarity of conditional vectors $\vec{c} = y + t$ across 1000 ImageNet classes using DiT-XL (Peebles & Xie, 2023).

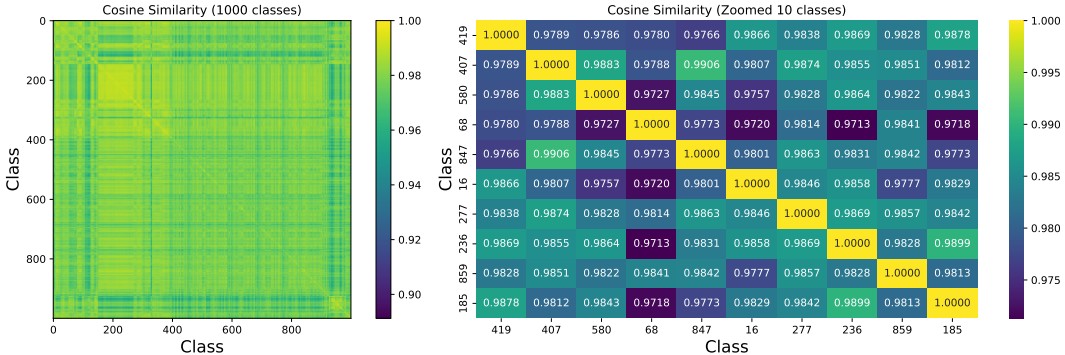

Figure 15: Cosine similarity of conditional vectors $\vec{c} = y + t$ across 1000 ImageNet classes using LightningDiT-XL (Yao et al., 2025).

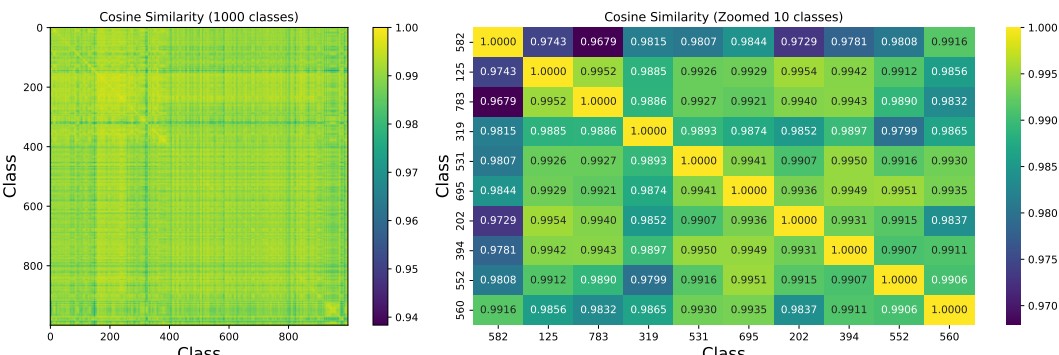

Figure 16: Cosine similarity of conditional vectors $\vec{c} = y + t$ across 1000 ImageNet classes using MDT-XL (Gao et al., 2023).

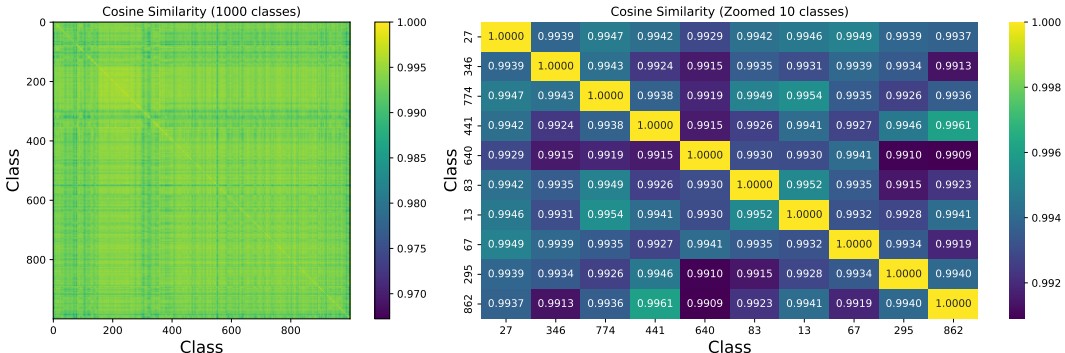

Figure 17: Cosine similarity of conditional vectors $\vec{c} = y + t$ across 1000 ImageNet classes using MG-XL (Tang et al., 2025).

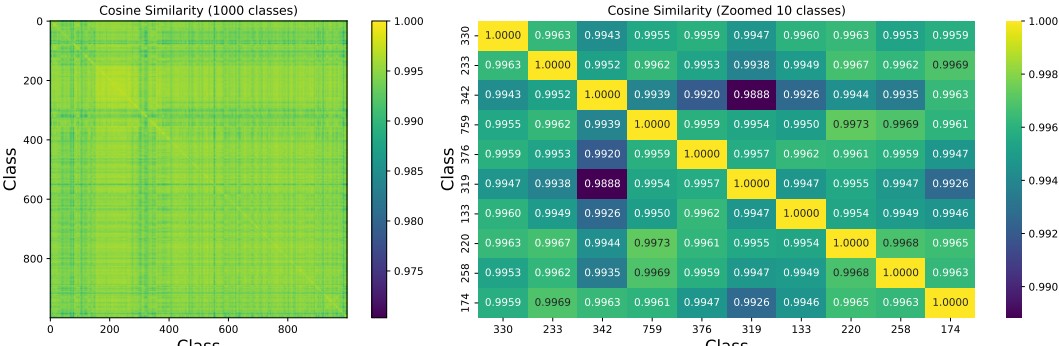

Figure 18: Cosine similarity of conditional vectors $\vec{c} = y + t$ across 1000 ImageNet classes using REPA-XL (Yu et al., 2025).

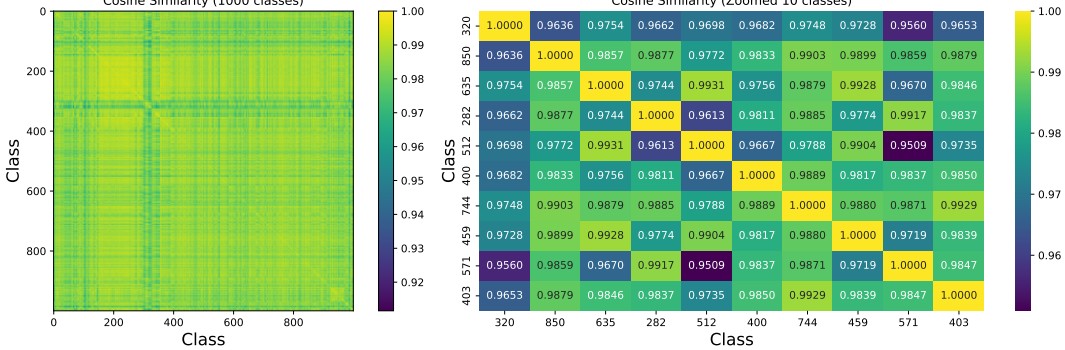

Figure 19: Cosine similarity of conditional vectors $\vec{c} = y + t$ across 1000 ImageNet classes using SiT-XL (Peebles & Xie, 2023).

A.4.2 T-SNE DISTRIBUTION ANALYSIS.

To further examine the role of head and tail dimensions in conditional embeddings, we provide t-SNE visualizations for all evaluated methods under targeted perturbations. Specifically, we manipulate either the high-magnitude (head) or low-magnitude (tail) dimensions of the embeddings and observe how these changes affect the overall distribution of class representations for all 1,000 ImageNet classes.

These visualizations illustrate that removing or altering head dimensions strongly disrupts the separability of class clusters, while perturbing tail dimensions has minimal impact, highlighting the concentration of semantic information in a small subset of dimensions.

Results for each method are shown in Fig. 20 – Fig. 25, providing a comparative view of how different architectures encode and distribute semantic information in their conditional embeddings.

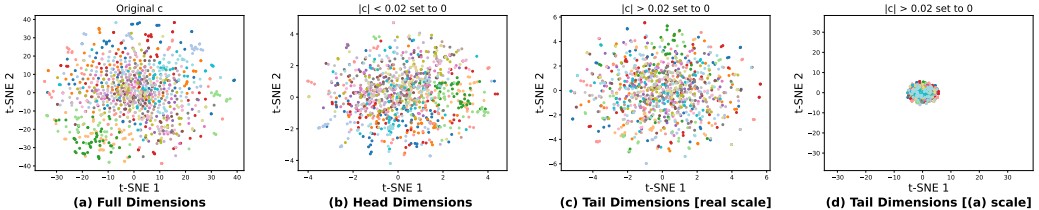

Figure 20: **(DiT) t-SNE of class embeddings by head vs. tail dimensions.** Keeping only head dimensions (b) preserves clear class clusters similar to the full embedding (a), while tail-only embeddings (c,d) collapse into entangled points, revealing weak semantic structure. Results are from DiT-XL on ImageNet-1K; similar trends appear in other models.

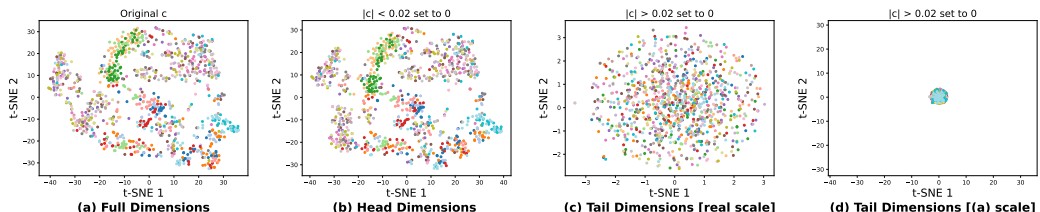

Figure 21: **(LightningDiT) t-SNE of class embeddings by head vs. tail dimensions.** Keeping only head dimensions (b) preserves clear class clusters similar to the full embedding (a), while tail-only embeddings (c,d) collapse into entangled points, revealing weak semantic structure. Results are from LightningDiT-XL on ImageNet-1K; similar trends appear in other models.

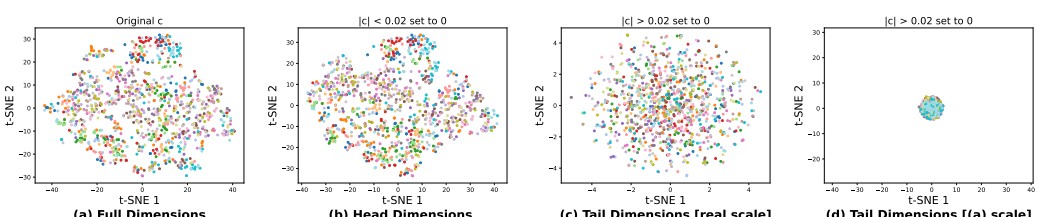

Figure 22: **(MDT) t-SNE of class embeddings by head vs. tail dimensions.** Keeping only head dimensions (b) preserves clear class clusters similar to the full embedding (a), while tail-only embeddings (c,d) collapse into entangled points, revealing weak semantic structure. Results are from MDT-XL on ImageNet-1K; similar trends appear in other models.

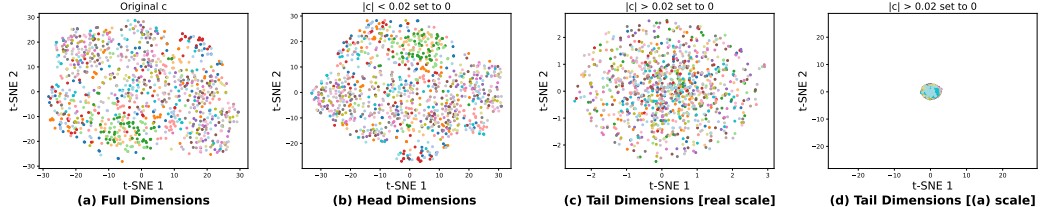

Figure 23: **(MG) t-SNE of class embeddings by head vs. tail dimensions.** Keeping only head dimensions (b) preserves clear class clusters similar to the full embedding (a), while tail-only embeddings (c,d) collapse into entangled points, revealing weak semantic structure. Results are from MG-XL on ImageNet-1K; similar trends appear in other models.

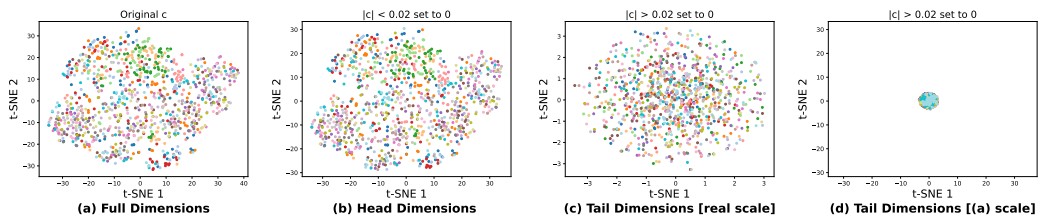

Figure 24: **(REPA) t-SNE of class embeddings by head vs. tail dimensions.** Keeping only head dimensions (b) preserves clear class clusters similar to the full embedding (a), while tail-only embeddings (c,d) collapse into entangled points, revealing weak semantic structure. Results are from REPA-XL on ImageNet-1K; similar trends appear in other models.

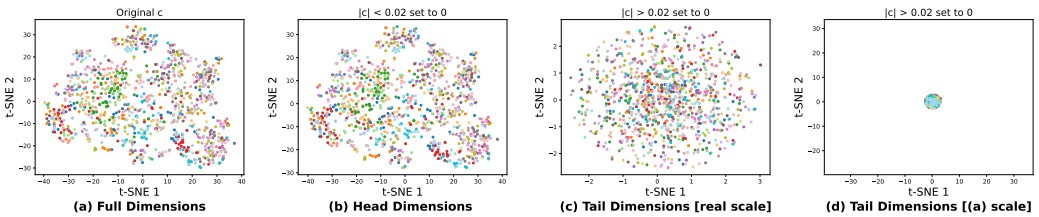

Figure 25: **(SiT) t-SNE of class embeddings by head vs. tail dimensions.** Keeping only head dimensions (b) preserves clear class clusters similar to the full embedding (a), while tail-only embeddings (c,d) collapse into entangled points, revealing weak semantic structure. Results are from SiT-XL on ImageNet-1K; similar trends appear in other models.

### A.4.3 Additional Baselines with Sparse Conditioning

We extend the evaluation from the main paper to two additional strong baselines: LightningDiT (Yao et al., 2025) and MG (Tang et al., 2025), following the same experimental protocol. As shown in Tab. 3, pruning low-magnitude dimensions in the conditional vector consistently improves both FID and CLIP scores. These results reinforce our main finding that dense conditional embeddings contain noisy, low-utility dimensions and that sparsification can yield more efficient and effective generative models.

Table 3: **More baselines.** Performance and semantic metrics under sparsification. $t_i$: prune every step, $t_0$: prune only at start, $t_{n-k,n}$: prune during last $k$ steps.

| Prune | Threshold $\tau$ | # Removed Dims | FID ↓ | IS ↑ | CLIP↑ |
|---|---|---|---|---|---|
| | Baseline MG (Tang et al., 2025) | 0/1152 (0%) | 7.2478 | 174.5151 | **30.199** |
| Tail | $\tau = 0.01$ ($t_i$) | 448/1152 (38.94%) | 7.2791 | 170.55 | 30.140 |
| | $\tau = 0.01$ ($t_0$) | 448/1152 (38.94%) | **7.2466** | **174.5537** | **30.199** |
| | $\tau = 0.01$ ($t_{n-k,n}$) | 448/1152 (38.94%) | **7.2455** | 174.3103 | **30.198** |
| . | Baseline LightningDiT (Yao et al., 2025) | 0/1152 (0%) | 7.0802 | 169.8574 | 30.720 |
| Tail | $\tau = 0.01$ ($t_i$) | 448/1152 (38.94%) | **7.0130** | 166.0569 | 30.7045 |
| | $\tau = 0.01$ ($t_0$) | 448/1152 (38.94%) | **7.0712** | 169.9164 | **30.729** |
| | $\tau = 0.01$ ($t_{n-k,n}$) | 448/1152 (38.94%) | **7.0745** | **169.9236** | **30.729** |

### A.4.4 Variance Distribution Analysis.

We analyze the per-dimension variance of the conditional embeddings by first computing the mean vector for each method and then measuring the variance across classes for each dimension. As expected, high-magnitude dimensions (head dimensions) exhibit substantially higher variance than the low-magnitude (tail) dimensions, reinforcing the observation that semantic information is concentrated in the head.

An exception is DiT, where the conditional vectors have smaller absolute values (maximum around 0.8, compared to 4–8 for other models), resulting in a different variance pattern. These results, visualized in Fig. 26 to Fig. 31, provide further evidence of the head–tail structure and its connection to semantic encoding in diffusion transformer embeddings.

For continuous-condition tasks such as pose-guided person image generation and video-guided audio generation, the learned embeddings are noticeably less sparse, consistent with the higher participation-ratio scores reported in Tab. 1 of the main paper. Detailed variance and mean analyses for these tasks are provided in Fig. 33 and Fig. 32.

### A.5 Additional Qualitative Results

We present an extended set of qualitative results for both class-conditional image generation on ImageNet and pose-guided person image synthesis. These visualizations highlight the impact of pruning low-magnitude dimensions in the conditional embedding vector.

Across a wide range of samples, we observe that removing these tail dimensions often preserves the generation quality and, in some cases, even enhances visual fidelity or sharpness. This supports our main finding that semantic information is concentrated in a small subset of head dimensions, while the majority of the embedding space is redundant.

Representative examples are provided in Fig. 34 through Fig. 41, demonstrating consistent trends across different models, classes, and poses.

### A.6 Use of Large Language Models

Large Language Models (LLMs) were used solely as a writing-assistance tool to polish grammar and improve sentence clarity. All research ideas, experimental design, analyses, and results were conceived and executed entirely by the authors. The LLM did not contribute to research ideation, data analysis, or the generation of any scientific content.

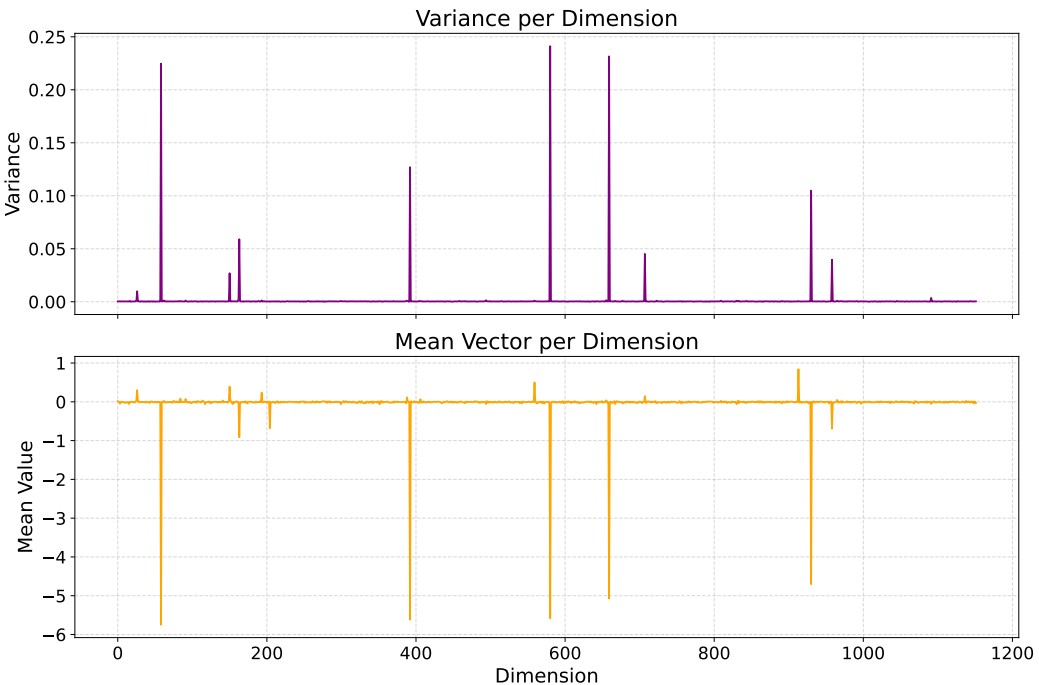

Figure 26: Variance per dimension of the conditional vector learned by MDT-XL.

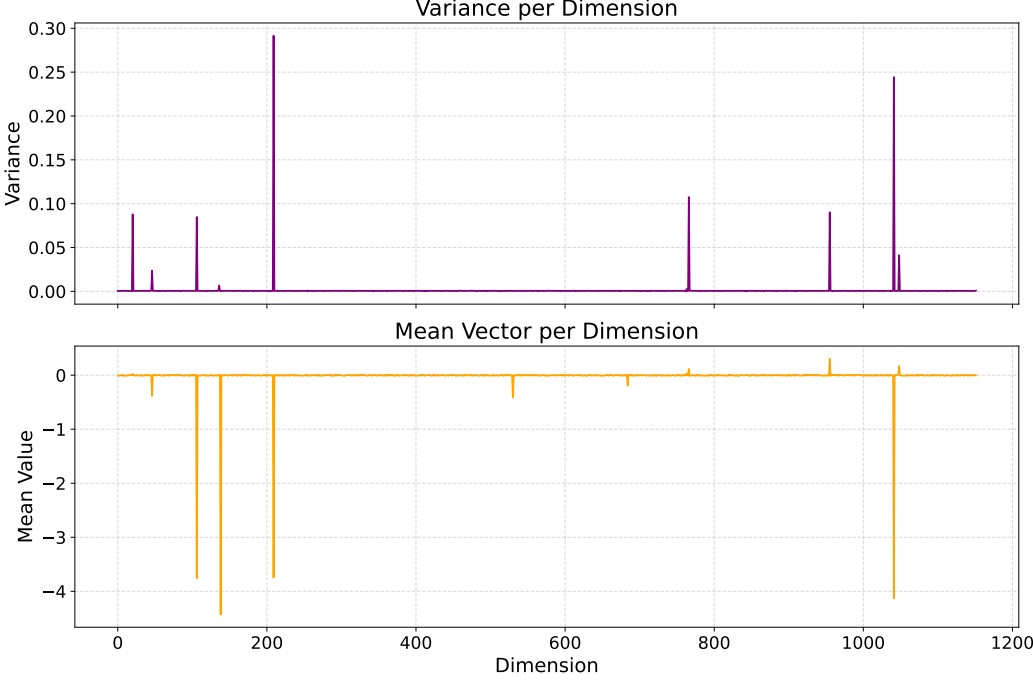

Figure 27: Variance per dimension of the conditional vector learned by LightningDiT-XL.

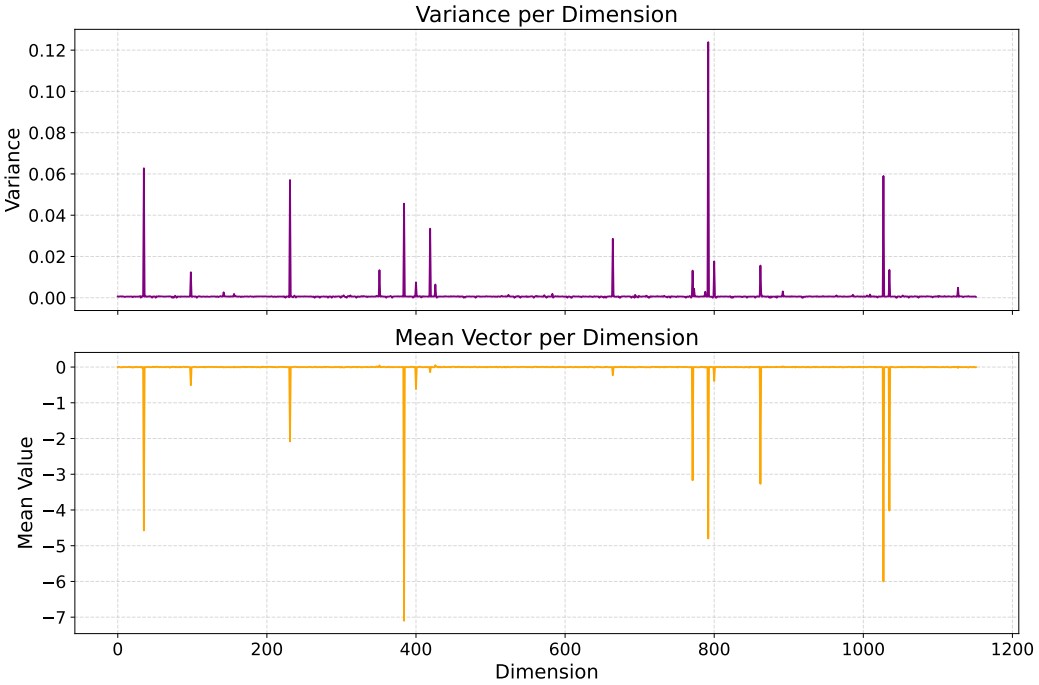

Figure 28: Variance per dimension of the conditional vector learned by MG-XL.

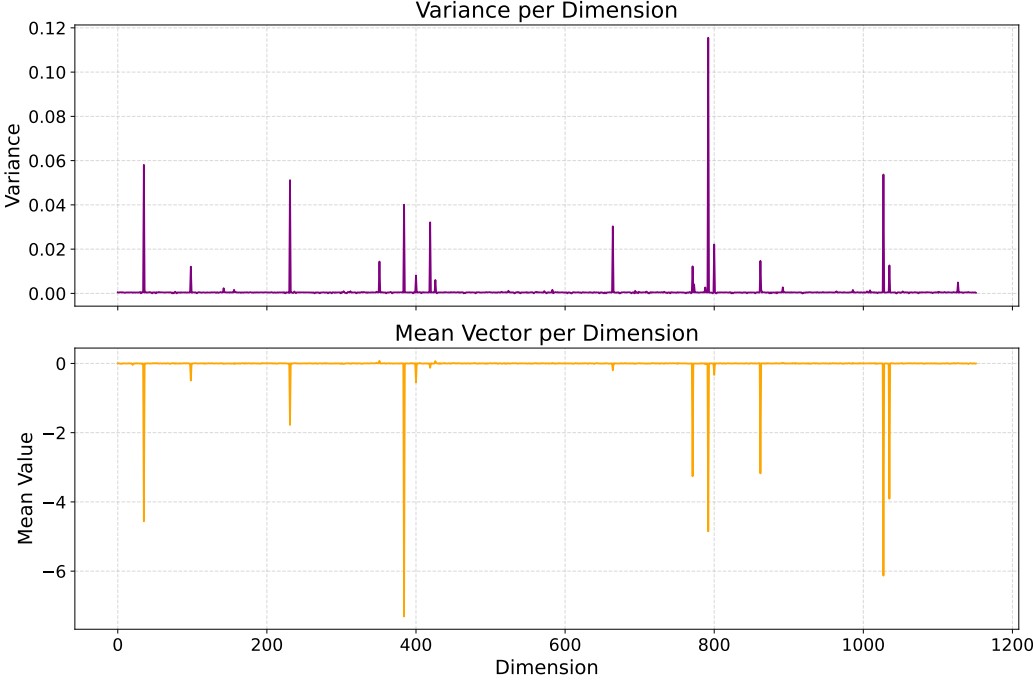

Figure 29: Variance per dimension of the conditional vector learned by REPA-XL.

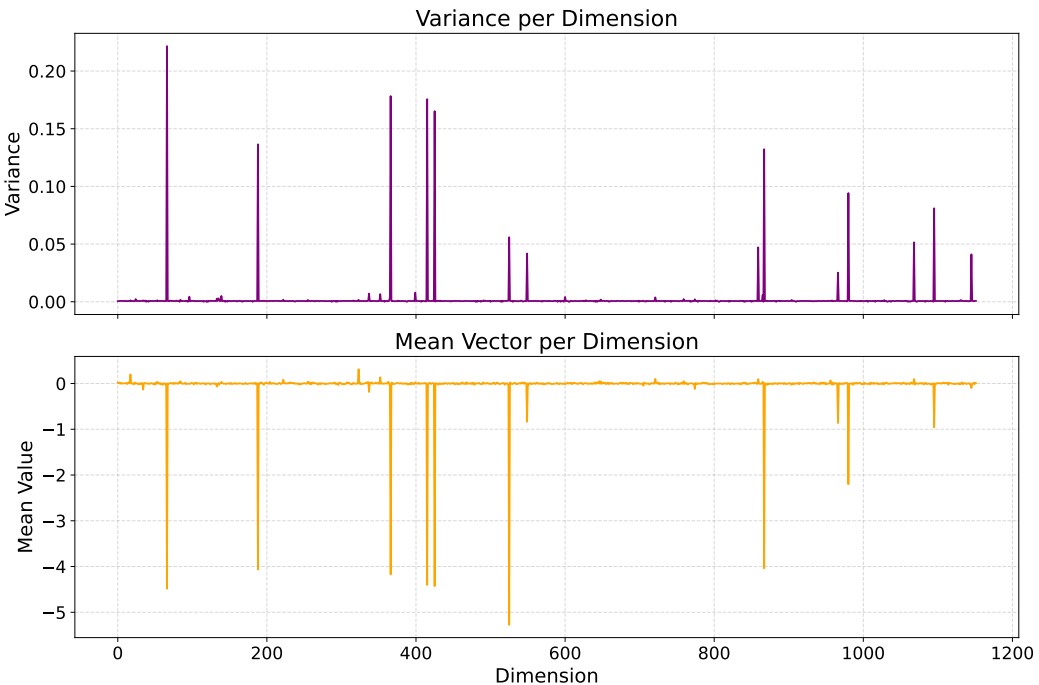

Figure 30: Variance per dimension of the conditional vector learned by SiT-XL.

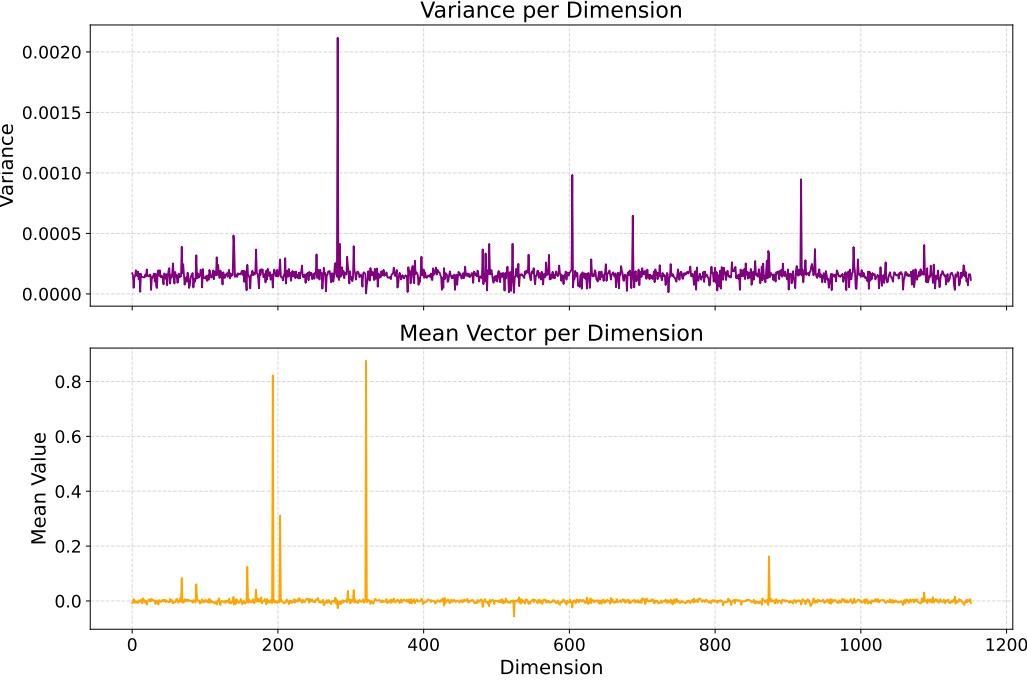

Figure 31: Variance per dimension of the conditional vector learned by DiT-XL.

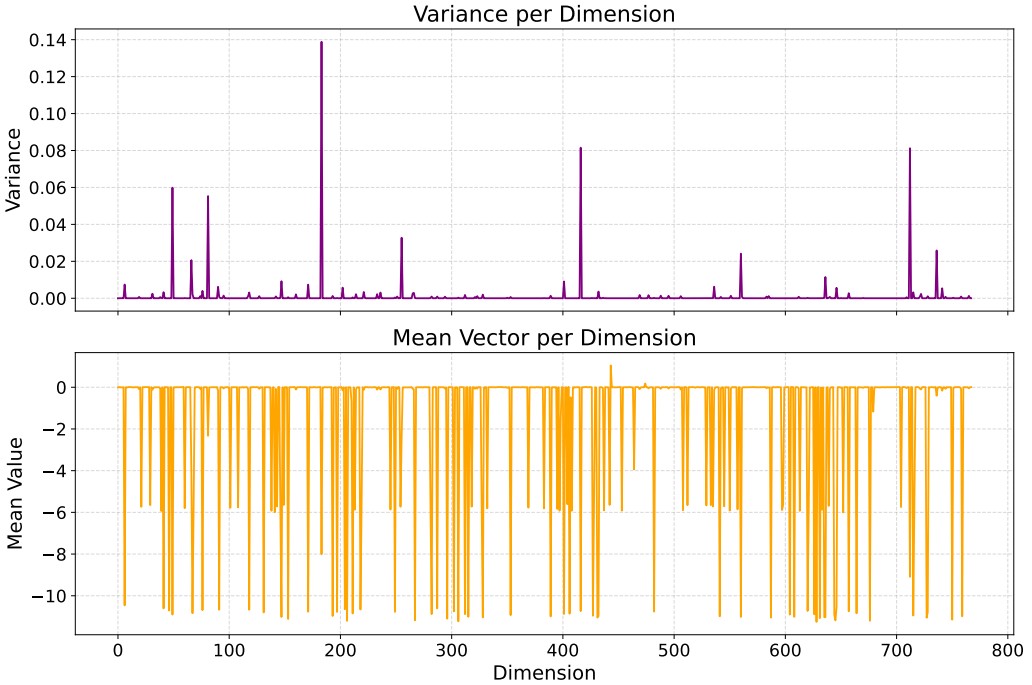

Figure 32: Variance per dimension of the conditional vector learned by MDSGen.

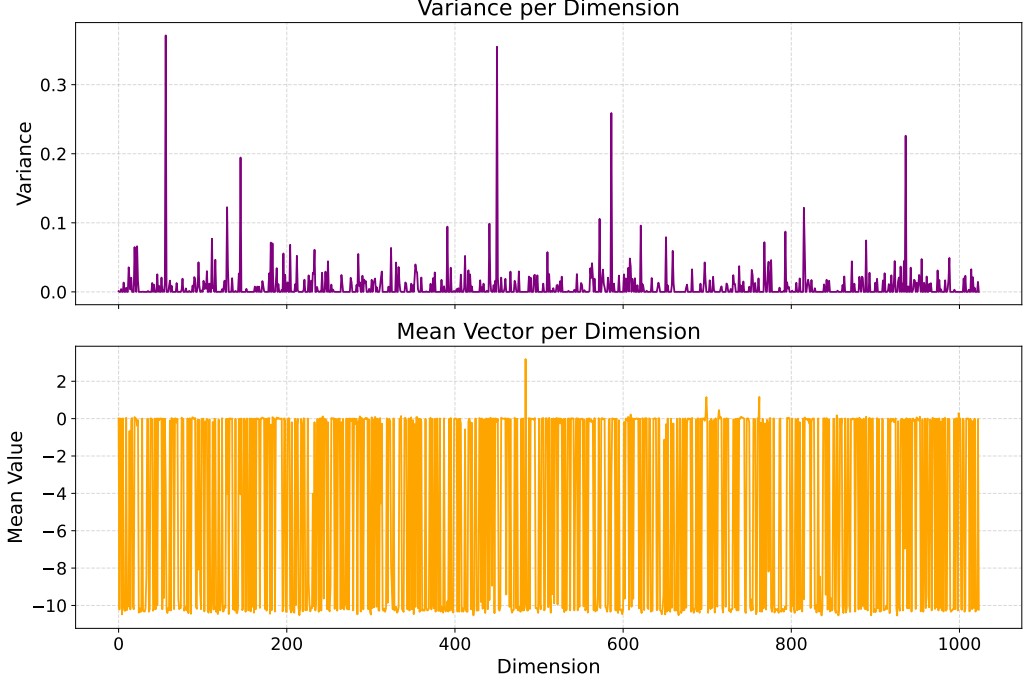

Figure 33: Variance per dimension of the conditional vector learned by X-MDPT.

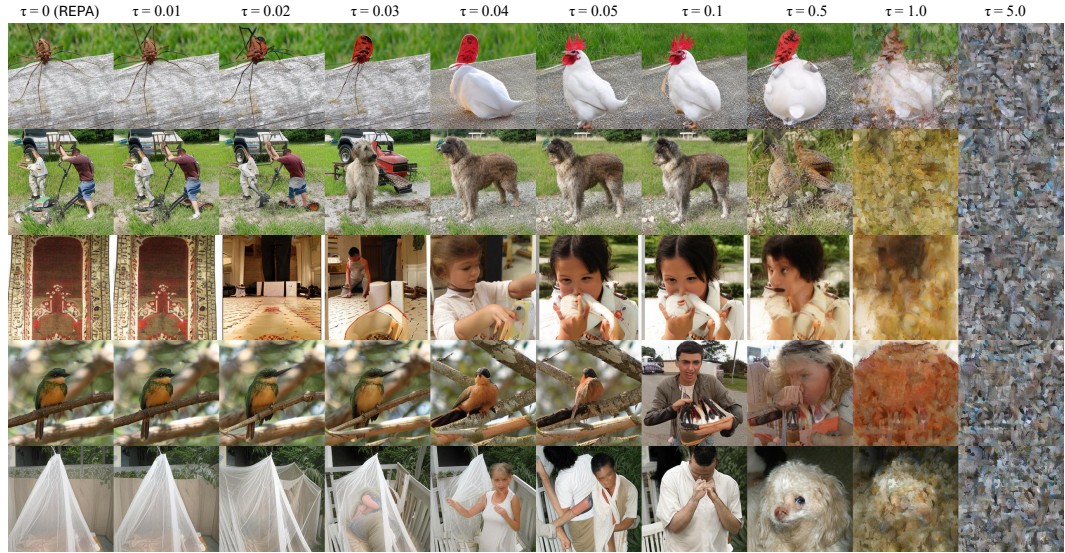

Figure 34: **Class-conditional ImageNet generation with pruned embeddings (1).** Removing low-magnitude dimensions from $\vec{c}$ preserves or slightly improves image quality, confirming that semantic information is concentrated in a few head dimensions.

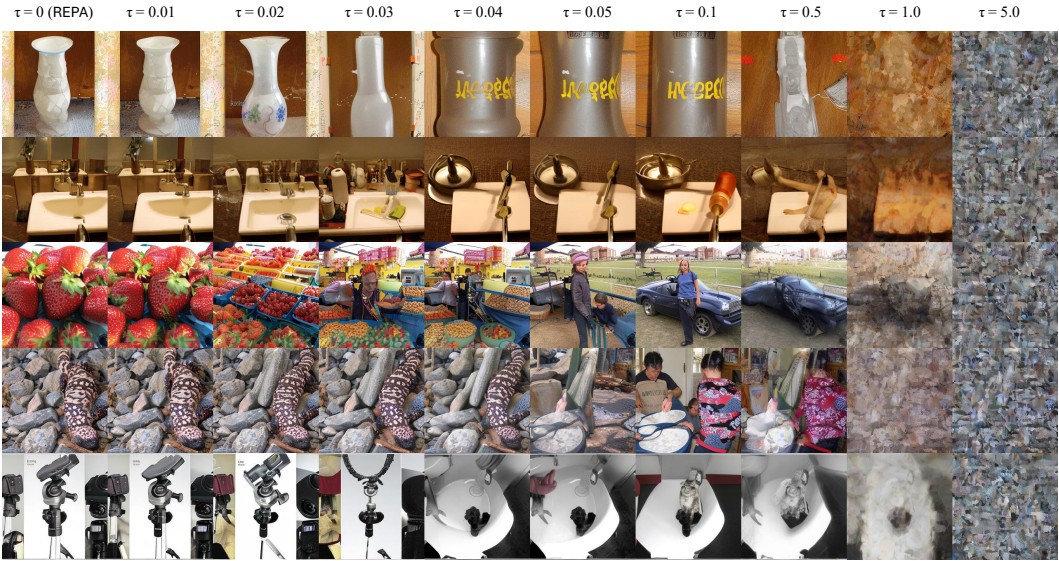

Figure 35: **Class-conditional ImageNet generation with pruned embeddings (2).** Removing low-magnitude dimensions from $\vec{c}$ preserves or slightly improves image quality, confirming that semantic information is concentrated in a few head dimensions.

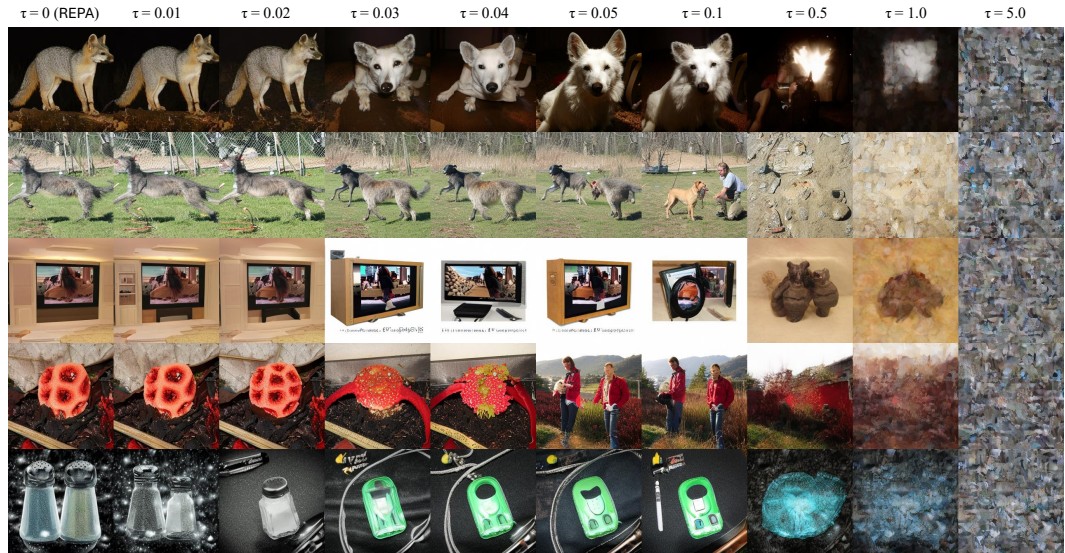

Figure 36: **Class-conditional ImageNet generation with pruned embeddings (3).**
Removing low-magnitude dimensions from $\vec{c}$ preserves or slightly improves image quality,
confirming that semantic information is concentrated in a few head dimensions.

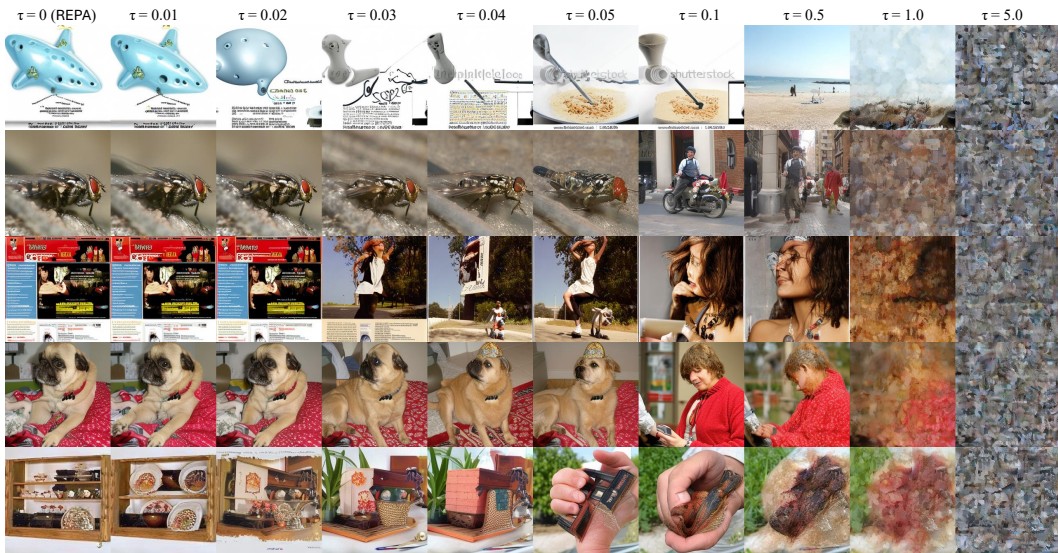

Figure 37: **Class-conditional ImageNet generation with pruned embeddings (4).**
Removing low-magnitude dimensions from $\vec{c}$ preserves or slightly improves image quality,
confirming that semantic information is concentrated in a few head dimensions.

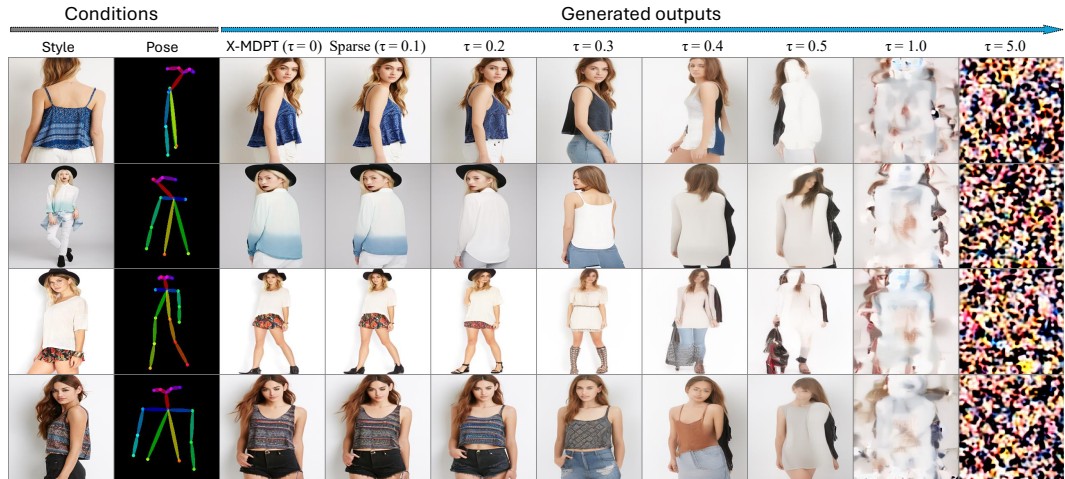

Figure 38: **Pose-guided person image synthesis with pruned embeddings (1).** Pruning tail dimensions in $\vec{c}$ maintains pose fidelity and visual quality, highlighting the redundancy of low-magnitude embedding dimensions.

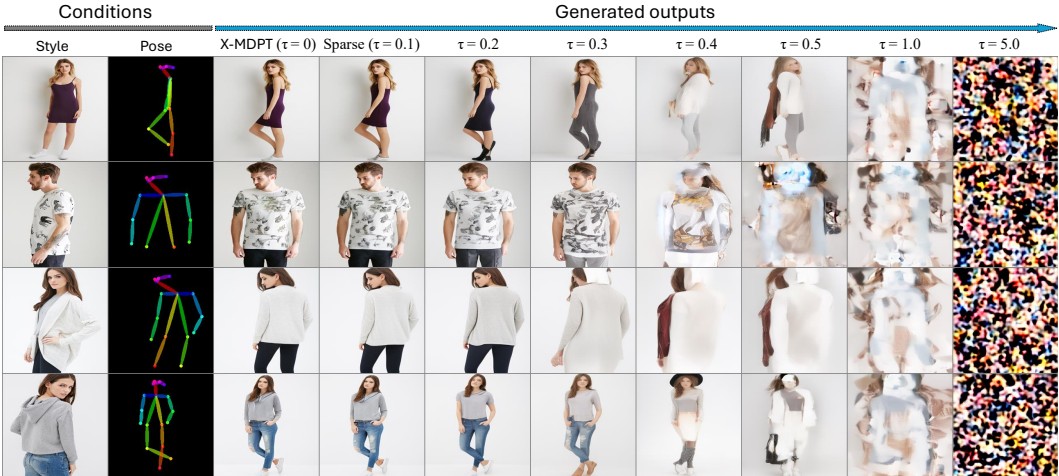

Figure 39: **Pose-guided person image synthesis with pruned embeddings (2).** Pruning tail dimensions in $\vec{c}$ maintains pose fidelity and visual quality, highlighting the redundancy of low-magnitude embedding dimensions.

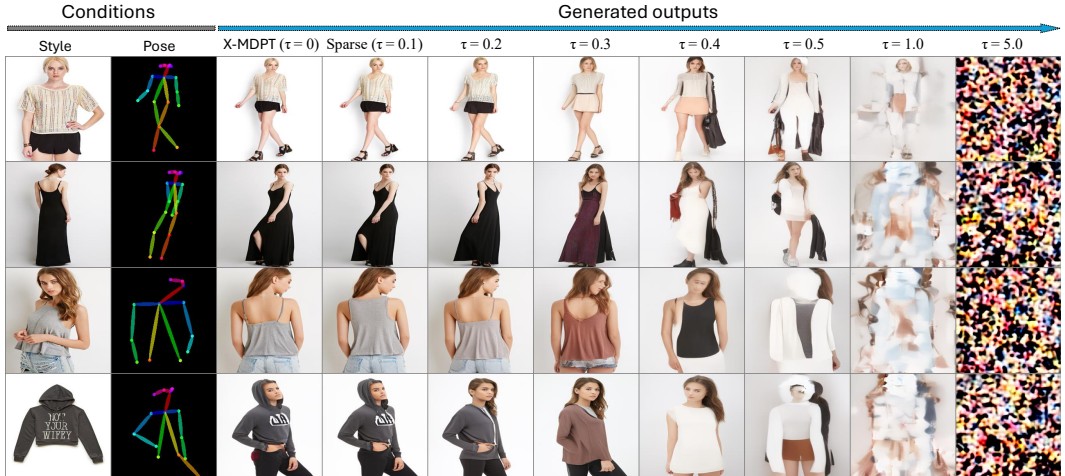

Figure 40: **Pose-guided person image synthesis with pruned embeddings (3).** Pruning tail dimensions in $\vec{c}$ maintains pose fidelity and visual quality, highlighting the redundancy of low-magnitude embedding dimensions.

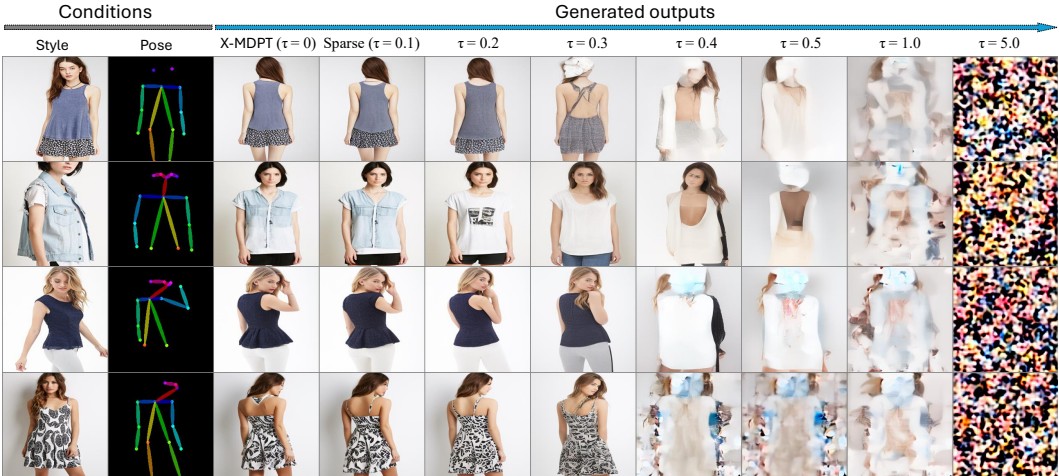

Figure 41: **Pose-guided person image synthesis with pruned embeddings (4).** Pruning tail dimensions in $\vec{c}$ maintains pose fidelity and visual quality, highlighting the redundancy of low-magnitude embedding dimensions.

# B  MORE BASELINES AND ANALYSIS

## B.1  Separating Timestep Embedding and Conditions Analysis.

Table 4: **Separate timestep (t) and conditions (y)**. Participation Ratio (PR) in learned conditional embeddings of state-of-the-art models on Imagenet-1K class-conditioned generation. With [1] denotes the methods used: **AdaLN**, and [2] denotes the method used: **concatenation**.

| Metrics | DiT[1] | SiT[1] | MDT[1] | LightningDiT[1] | MG[1] | REPA[1] | UViT[2] | Embed. |
|---|---|---|---|---|---|---|---|---|
| Cosine Sim. | **0.9001** | **0.9852** | **0.9905** | **0.9779** | **0.9934** | **0.9946** | **0.97917** | y + t |
| nPR ($\alpha_{norm}$) % | 10.47 | 2.28 | 1.60 | 2.05 | 1.73 | 1.43 | 50.06 | y + t |
| Cosine Sim. | 0.7774 | 0.5436 | 0.8540 | 0.7166 | 0.6853 | 0.5194 | 0.00165 | y |
| nPR ($\alpha_{norm}$) % | 70.14 | 37.43 | 36.75 | 36.42 | 43.67 | 41.60 | 63.52 | y |

## B.2  Text-conditioned Methods and Model Sizes.

Table 5: **Separate timestep (t) and conditions (y)**. Participation Ratio (PR) in learned conditional embeddings of state-of-the-art models on text or video-conditioned generation. With [1] denotes the methods used: **AdaLN**, and [3] denotes the method used: **cross-attention**.

| Metrics | X-MDPT-L[1] | X-MDPT-B[1] | X-MDPT-S[1] | SD3.0 (2B)[1] | SD3.0 (8B)[1] | MDSGen[1] | AudioLDM[3] | Embed. |
|---|---|---|---|---|---|---|---|---|
| Cosine Sim. | **0.9998** | **0.99992** | **0.9995** | **0.9962** | **0.9995** | **0.9999** | **0.9828** | y + t |
| nPR ($\alpha_{norm}$) % | 48.42 | 37.59 | 53.41 | 54.79 | 26.67 | 13.57 | 8.62 | y + t |
| Cosine Sim. | **0.9862** | **0.9909** | **0.9492** | **0.9949** | **0.9937** | **0.9918** | 0.1406 | y |
| nPR ($\alpha_{norm}$) % | 20.68 | 29.75 | 37.51 | 52.39 | 24.25 | 9.98 | 63.09 | y |

## B.3  More Quantitative Metrics.

Table 6: **Precision and Recall** with previous metrics: FID, IS, and CLIP.

| Method | FID↓ | IS↑ | CLIP↑ | Precision↑ | Recall↑ | Remark |
|---|---|---|---|---|---|---|
| REPA (Yu et al., 2025) | 7.1694 | **176.02** | 29.746 | 0.8032 | 0.6236 | Baseline |
| Pruned ($\tau = 0.01$) $t_0$ | **7.1690** | 175.97 | **29.807** | 0.7878 | **0.6252** | **Ours** |
| Pruned ($\tau = 0.01$) $t_{n-k,n}$ | **7.1598** | 175.49 | **29.805** | **0.8045** | **0.6381** | **Ours** |
| Model-Guide (Tang et al., 2025) | 7.2478 | 174.5151 | **30.199** | 0.7842 | 0.6633 | Baseline |
| Pruned ($\tau = 0.01$) $t_0$ | **7.2466** | **174.5537** | **30.199** | **0.7854** | 0.6625 | **Ours** |
| Pruned ($\tau = 0.01$) $t_{n-k,n}$ | **7.2455** | 174.3103 | 30.198 | **0.7898** | **0.6644** | **Ours** |
| LightningDiT (Yao et al., 2025) | 7.0802 | 169.8574 | 30.720 | 0.7928 | 0.6248 | Baseline |
| Pruned ($\tau = 0.01$) $t_0$ | **7.0712** | 169.9164 | **30.729** | 0.7906 | **0.6256** | **Ours** |
| Pruned ($\tau = 0.01$) $t_{n-k,n}$ | **7.0745** | **169.9236** | **30.729** | **0.7935** | **0.6265** | **Ours** |

Table 7: Quantitative metrics on the DeepFashion dataset of pose-guide person image generation task with masked diffusion transformers.

| Method | FID ↓ | SSIM↑ | LPIPS↓ | PSNR↑ | Remark |
|---|---|---|---|---|---|
| X-MDPT (Pham et al., 2024) | 18.6372 | 0.6798 | 0.1672 | 17.336 | Baseline |
| Pruned ($\tau = 0.1$) 40% | 18.6692 | 0.6792 | 0.1675 | 17.328 | **Ours** |

## B.4  Computational Reduction Analysis.

We provide the benefit of sparse embedding when pruning in Fig. 42.

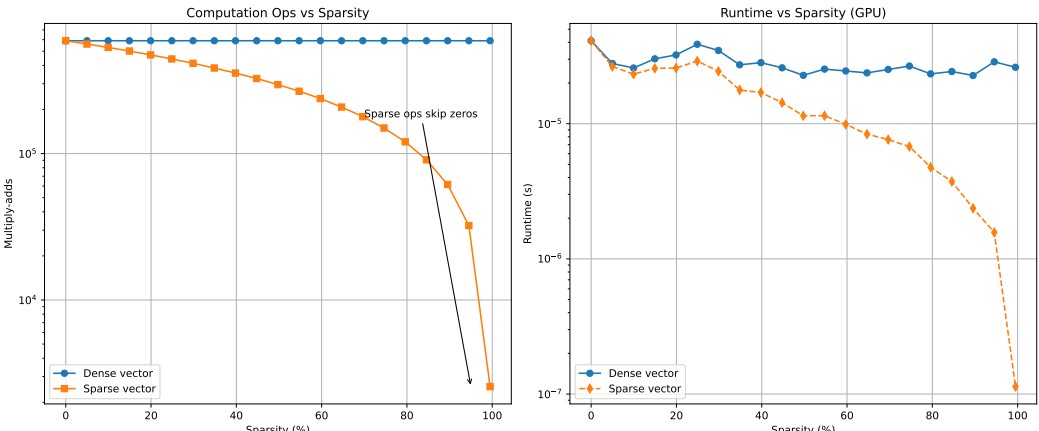

Figure 42: **Dense vs. Sparse vectors.** Compared the computation overhead. It shows that a sparse vector is more efficient in computation and has faster runtime than a dense vector (baseline).

## B.5 Stability Analysis with Limited Timestep.

Durign the rebuttal, the reviewer asked for the generalization of limited timestep embedding. We perform a more comprehensive experiment in Fig. 43.

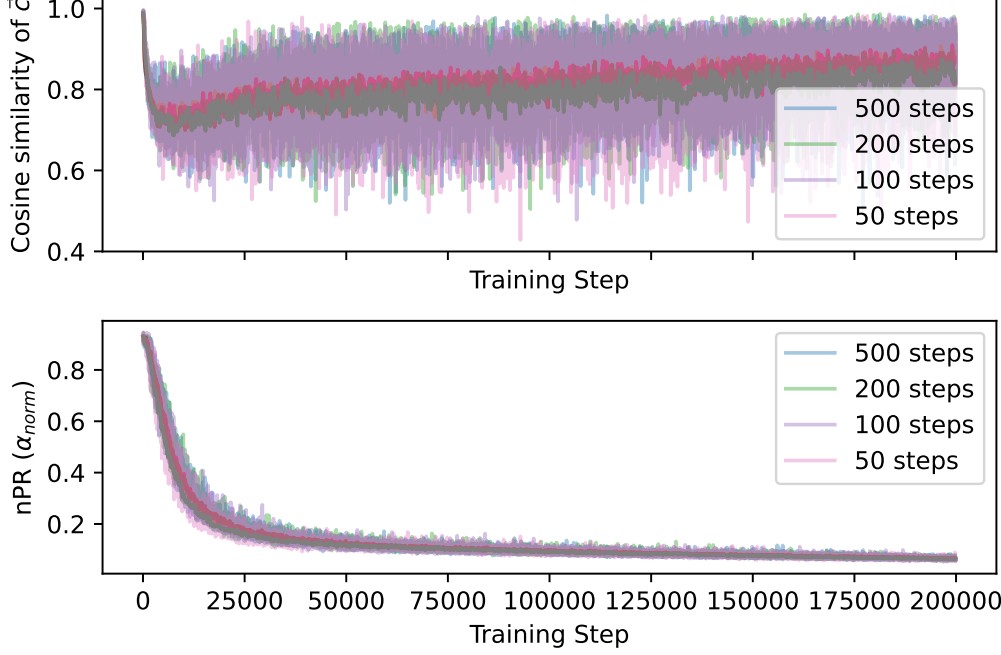

Figure 43: **Training dynamics of conditional embeddings.** Top: batchwise cosine similarity of $\vec{c}$ during training. Bottom: participation ratio (nPR) over training steps, showing progressive sparsification. **With limited timesteps: 50, 100, 200, and 500.**

## B.6 Additional Related Work

### B.6.1 Multimodal Representation and Embedding Space

In this section, we provide a broader overview of foundational techniques in multimodal representation learning that inform our semantic bottleneck design. Early works explored diverse strategies for modality alignment and flexible fusion and data augmentation techniques, as well as in vision and speech processing (Lee et al., 2020; Jung et al., 2020; 2022; Vu et al., 2019; Trung & Yoo, 2019). Specifically, techniques focusing on modality-specific constraints (Kim et al., 2020) and self-supervised alignment (Pham et al., 2021; 2023) laid the groundwork for establishing robust embedding spaces. Recent advances have further refined these representations through deep architectural optimizations, which are essential for maintaining semantic integrity within the conditioning signals of diffusion models.

### B.6.2 Generative Modeling and Diffusion Refinement

The integration of Transformers into diffusion frameworks has shifted the focus toward scalable and semantically aware conditioning. Recent studies, such as MDSGen (Pham et al., 2025b) and MD3C (Pham et al., 2025a), demonstrate the efficacy of complex multi-data scheduling and contrastive constraints in generative tasks. Additionally, exploring the capabilities of these models in specialized domains (Yoon et al., 2025; Hong et al., 2025; Pham et al., 2024; Koo et al., 2024; 2025) and structural adaptation (Pham et al., 2022b;a) provides a comprehensive context for our proposed Diffusion Transformer with semantic bottlenecking.

