# OpenReview forum: "A Hidden Semantic Bottleneck in Conditional Embeddings of Diffusion Transformers"
_ICLR.cc/2026/Conference — ICLR 2026 Poster_

### Official Review · Reviewer_TVaD · 2025-10-29

**Soundness:** 2
**Presentation:** 2
**Contribution:** 3
**Rating:** 6
**Confidence:** 3

**Summary:**

This paper investigates the properties of conditional embeddings in conditional diffusion models, revealing that they are sparse yet locally uniform across dimensions. The analysis is conducted on several representative class-conditional diffusion models and is extendable to continuous class embeddings and diverse conditioning signals, e.g., video-to-audio and pose-guided image generation. By removing redundant dimensions, slight improvements in FID and CLIP scores are observed.

**Strengths:**

- The finding that conditional embeddings in DiT are sparse but locally uniform is meaningful, as it reveals redundancy in the conditional representations of current conditional diffusion models. This insight can motivate new directions for future research.
- Both qualitative and quantitative experiments on diffusion models with various types of conditions validate that conditional embeddings in current DiTs exhibit significant redundancy.
- The authors demonstrate that simply removing redundant dimensions in conditional embeddings at inference time can enhance generation quality to some extent, highlighting the potential of optimizing conditional embeddings to improve DiTs.

**Weaknesses:**

- The proposed hypotheses lack experimental support. For example, regarding the hypothesis about the cause of high cosine similarity in conditional embeddings, the authors argue that DiTs favor embeddings providing stable and robust signals for denoising because they condition on embeddings across all timesteps. Maybe a training log similar to Fig. 12, but for a model trained on fewer timesteps, could validate this hypothesis. Since inference performance is not the concern here, such a model could be trained on a subset of timesteps.
- There is a contradiction between Lines 454–457, which state that high cosine similarity occurs only in transformers and not in U-Nets, and Lines 467–470, which state that similar redundancy can also appear in U-Net-based diffusion models. Moreover, the authors neither present nor discuss experimental results on U-Net diffusion models.
- The authors claim that AdaLN amplifies high-magnitude dimensions of conditional embeddings, thereby preserving semantic differences, but this claim also lacks experimental validation. What happens if AdaLN is replaced with another conditioning mechanism, such as cross-attention?

**Questions:**

What is the setting of k in Table 2?

---

> ### Author Response · Authors · 2025-11-25
> **Response 1/N**
>
> Dear Reviewer **TVaD**,
>
> Thank you for your thoughtful and constructive feedback. We appreciate your comments and address each of your concerns point by point below:
>
> > **W1.** The proposed hypotheses lack experimental support. For example, regarding the hypothesis about the cause of high cosine similarity in conditional embeddings, the authors argue that DiTs favor embeddings providing stable and robust signals for denoising because they condition on embeddings across all timesteps. Maybe a training log similar to Fig. 12, but for a model trained on fewer timesteps, could validate this hypothesis. Since inference performance is not the concern here, such a model could be trained on a subset of timesteps.
>
> **Re:** As suggested, we conducted additional experiments using **fewer timesteps** to validate our hypothesis. While most SOTA models use (T = 1000) timesteps, we trained models with (T = 50, 100, 200,) and (500) and repeated the analysis from Fig. 12. The results (shown in the figure below) confirm that embeddings across fewer timesteps exhibit **similar trends in cosine similarity**, supporting the hypothesis that conditioning across all timesteps contributes to the observed redundancy.
>
> The results are included in the revised manuscript and Appendix, accessible via the anonymized link:
>
> **https://i.postimg.cc/hGvtVnpB/6-stability-with-limited-timestep.png**

---

> ### Author Response · Authors · 2025-11-25
> **Response 2/N**
>
> > **W2.** There is a contradiction between Lines 454–457, which state that high cosine similarity occurs only in transformers and not in U-Nets, and Lines 467–470, which state that similar redundancy can also appear in U-Net-based diffusion models. Moreover, the authors neither present nor discuss experimental results on U-Net diffusion models.
>
> **Re:** Thank you for pointing this out. Based on additional experiments with U-Net models, we clarify that **high cosine similarity occurs only in transformers and not in U-Nets when timestep embeddings are removed**. However, **similar redundancy can appear in U-Net–based diffusion models when timestep embeddings are included**. We have revised the manuscript to reflect this distinction.
>
> **Table 1: Separate timestep (t) and conditions (y)**
> Participation Ratio (PR) in learned conditional embeddings of state-of-the-art models on text or video-conditioned generation.
> `¹` denotes **AdaLN** and `³` denotes **cross-attention**.
>
> | **Metrics**        | **X-MDPT-L¹** | **X-MDPT-B¹** | **X-MDPT-S¹** | **SD3.0 (2B)¹** | **SD3.0 (8B)¹** | **MDSGen¹** | **AudioLDM³** | **Embed.** |
> | ------------------ | ------------- | ------------- | ------------- | --------------- | --------------- | ----------- | ------------- | ---------- |
> | **Cosine Sim.**    | **0.9998**    | **0.99992**   | **0.9995**    | **0.9962**      | **0.9995**      | **0.9999**  | **0.9828**    | y + t      |
> | **nPR (α_norm) %** | 48.42         | 37.59         | 53.41         | 54.79           | 26.67           | 13.57       | 8.62          | y + t      |
> | ---                | ---           | ---           | ---           | ---             | ---             | ---         | ---           | ---        |
> | **Cosine Sim.**    | **0.9862**    | **0.9909**    | **0.9492**    | **0.9949**      | **0.9937**      | **0.9918**  | 0.1406        | y          |
> | **nPR (α_norm) %** | 20.68         | 29.75         | 37.51         | 52.39           | 24.25           | 9.98        | 63.09         | y          |
>
> > **W3.** The authors claim that AdaLN amplifies high-magnitude dimensions of conditional embeddings, thereby preserving semantic differences, but this claim also lacks experimental validation. What happens if AdaLN is replaced with another conditioning mechanism, such as cross-attention?
>
> **Re:** We conducted experiments replacing AdaLN with alternative conditioning mechanisms—**cross-attention (AudioLDM)** and **concatenation (UViT)**. The results, shown in the table below, indicate that cross-attention and concatenation produce **substantially lower cosine similarity** (e.g., 0.14 for AudioLDM and 0.0016 for UViT with condition-only embeddings) compared to AdaLN-based approaches (REPA, SiT, Stable Diffusion 3.0, X-MDPT, MDSGen, etc.). This confirms that the observed extreme redundancy is **closely tied to the AdaLN mechanism** rather than a general property of all conditioning methods.
>
> **Table 2: Separate timestep (t) and conditions (y)**
> Participation Ratio (PR) in learned conditional embeddings of state-of-the-art models on Imagenet-1K class-conditioned generation.
> `¹` denotes **AdaLN** methods and `²` denotes **concatenation**.
>
> | **Metrics**        | **DiT¹**   | **SiT¹**   | **MDT¹**   | **LightningDiT¹** | **MG¹**    | **REPA¹**  | **UViT²**   | **Embed.** |
> | ------------------ | ---------- | ---------- | ---------- | ----------------- | ---------- | ---------- | ----------- | ---------- |
> | **Cosine Sim.**    | **0.9001** | **0.9852** | **0.9905** | **0.9779**        | **0.9934** | **0.9946** | **0.97917** | y + t      |
> | **nPR (α_norm) %** | 10.47      | 2.28       | 1.60       | 2.05              | 1.73       | 1.43       | 50.06       | y + t      |
> | ---                | ---        | ---        | ---        | ---               | ---        | ---        | ---         | ---        |
> | **Cosine Sim.**    | 0.7774     | 0.5436     | 0.8540     | 0.7166            | 0.6853     | 0.5194     | 0.00165     | y          |
> | **nPR (α_norm) %** | 70.14      | 37.43      | 36.75      | 36.42             | 43.67      | 41.60      | 63.52       | y          |
>
> > **Q1.** What is the setting of k in Table 2?
>
> **Re:** In Table 2 (in the main paper), (k) denotes the number of **final diffusion steps** during which pruning is applied. In our experiments, we set (k = 10), meaning pruning is performed only in the **last 10 steps** of the iterative inference process.
>
> ---
>
> Best regards,
>
> **Submission635 Authors**

---

### Official Review · Reviewer_jVCn · 2025-10-31

**Soundness:** 4
**Presentation:** 4
**Contribution:** 2
**Rating:** 4
**Confidence:** 4

**Summary:**

This is a very interesting work. This paper makes significant contributions to the understanding of conditional embeddings in diffusion Transformers, a critical component of state-of-the-art generative models.

**Strengths:**

1. It is the first work to systematically investigate the internal structure of conditional embeddings in diffusion Transformers, uncovering two core emergent properties—extreme angular similarity (exceeding 99% on ImageNet-1K and 99.9% on continuous-condition tasks) and high-dimensional sparsity (only 1–2% of dimensions carrying substantial semantic information). This fills a critical gap in existing literature, which has primarily focused on architectural advances rather than the intrinsic characteristics of conditional encoding.
2. The experiments are well-designed and extensive, covering six diffusion Transformer models (DiT, MDT, SiT, REPA, LightningDiT, MG) on discrete class-conditional image generation (ImageNet-1K) and two continuous-condition tasks (pose-guided image synthesis with X-MDPT and video-to-audio generation with MDSGen).
3. The demonstration that pruning up to 66% of low-magnitude dimensions preserves or even improves generation quality provides actionable insights for model optimization. This offers a potential path toward more efficient conditioning mechanisms—particularly valuable for resource-constrained applications.

**Weaknesses:**

1. Although the author reveals the redundancy problem in condition embeddings of diffusion models via empirical analysis, they do not provide any further method to improve the efficiency or performance of existing models based on the observation.
2. More conditions should be considered. For instance, user-generated descriptions or prompts are more general than class-conditional prompts. I suggest the author investigate more types of conditions.
3. Maybe the time embedding and prompt embedding should be considered separately.

**Questions:**

see weaknesses

---

> ### Author Response · Authors · 2025-11-25
> **Response 1/N**
>
> Dear Reviewer **jVCn**,
>
> We sincerely appreciate your review and constructive comments. Below, we provide detailed responses to each of your points:
>
> > **W1.** Although the author reveals the redundancy problem in condition embeddings of diffusion models via empirical analysis, they do not provide any further method to improve the efficiency or performance of existing models based on the observation.
>
> **Re:** Our primary goal in this work is to **identify, analyze, and characterize** the structural redundancy in conditional embeddings of diffusion models. While developing new methods to exploit this phenomenon is an important future direction, it is beyond the current scope. Nonetheless, our findings already provide actionable insights: pruning redundant dimensions produces genuinely **sparse conditional vectors**, which can be leveraged to reduce computation. As shown in the figure below, sparse vectors achieve **faster runtime and lower computational cost** compared to dense vectors, demonstrating that the discovered redundancy can directly inform more efficient system design. We will clarify in the revised manuscript that the main contribution lies in **revealing and explaining the phenomenon**, while the development of new optimization or architectural techniques is left for future work.
>
> The results are included in the revised manuscript and accessible via the anonymized link:
>
> **https://i.postimg.cc/cLsH4w3Z/5-sparse-vs-dense-gpu-fullrange-rebut-dim-1152.png**
>
>
> > **W2.** More conditions should be considered. For instance, user-generated descriptions or prompts are more general than class-conditional prompts. I suggest the author investigate more types of conditions.
>
> **Re:** As suggested, we extended our analysis to more general conditioning types, including **user-generated prompts** in **text-to-image** (Stable Diffusion v3.0) and **text-to-audio** (AudioLDM) models. Consistent with our original findings, these models also exhibit **high redundancy** in conditional embeddings when timestep embeddings are included, while alternative injection schemes show reduced redundancy when timestep embeddings are excluded. The results are summarized in the figure below and will be added to the Appendix.
>
>
> **Table 1: Separate timestep (t) and conditions (y)**
> Participation Ratio (PR) in learned conditional embeddings of state-of-the-art models on text or video-conditioned generation.
> `¹` denotes **AdaLN** and `³` denotes **cross-attention**.
>
> | **Metrics**        | **X-MDPT-L¹** | **X-MDPT-B¹** | **X-MDPT-S¹** | **SD3.0 (2B)¹** | **SD3.0 (8B)¹** | **MDSGen¹** | **AudioLDM³** | **Embed.** |
> | ------------------ | ------------- | ------------- | ------------- | --------------- | --------------- | ----------- | ------------- | ---------- |
> | **Cosine Sim.**    | **0.9998**    | **0.99992**   | **0.9995**    | **0.9962**      | **0.9995**      | **0.9999**  | **0.9828**    | y + t      |
> | **nPR (α_norm) %** | 48.42         | 37.59         | 53.41         | 54.79           | 26.67           | 13.57       | 8.62          | y + t      |
> | ---                | ---           | ---           | ---           | ---             | ---             | ---         | ---           | ---        |
> | **Cosine Sim.**    | **0.9862**    | **0.9909**    | **0.9492**    | **0.9949**      | **0.9937**      | **0.9918**  | 0.1406        | y          |
> | **nPR (α_norm) %** | 20.68         | 29.75         | 37.51         | 52.39           | 24.25           | 9.98        | 63.09         | y          |

---

> > ### Author Response · Authors · 2025-11-25
> > **Response 2/N**
> >
> > > **W3.** Maybe the time embedding and prompt embedding should be considered separately.
> >
> > **Re:** To isolate the effects, we conducted a **side-by-side analysis** of the conditional embeddings and timestep embeddings separately. The results, shown in the figure below, reveal the individual contributions of each component to redundancy and participation rate, providing a clearer understanding of how they shape the overall embedding structure.
> >
> > For example (Table 1 above), in **Stable Diffusion v3.0** evaluated on the COCO validation set, two semantically distinct captions—“A group of people shopping at an outdoor market” and “A skier is racing swiftly down the mountain”—produce embeddings that are already **99.37% similar** after the learnable projection layers. Adding the timestep embedding further increases similarity to **99.95%**. This finding is consistent with observations across other baselines in our submission, highlighting that both embeddings contribute to the high redundancy phenomenon.
> >
> > ---
> >
> > Best regards,
> >
> > **Submission635 Authors**

---

> > ### Comment · Reviewer_jVCn · 2025-11-26
> > **Response to rebuttal**
> >
> > Thanks for your response. I still have concerns about W1. In Table 2 of the main paper, removing  66.21% of the tail embedding elements slightly hinders generation quality. In this figure <https://i.postimg.cc/cLsH4w3Z/5-sparse-vs-dense-gpu-fullrange-rebut-dim-1152.png>, I notice that the FLOPs are only reduced by about 50% when removing 60% of the tail embedding elements. Thus, a natural question arises: compared to existing distilling-based models, e.g., SD-Flash, the decrease in computational cost is not remarkable. Therefore, I still insist on my grading.

---

> > > ### Author Response · Authors · 2025-11-28
> > >
> > > Dear Reviewer **jVCn**,
> > >
> > > Thank you very much for your follow-up and for revisiting our work. We appreciate your continued concern regarding **W1**. Your point about the limited computational reduction compared to distillation-based approaches is well noted.
> > >
> > > Our focus in this paper is to identify and analyze the redundancy phenomenon rather than to replace distillation, but your comment highlights an important direction that we will clarify and further emphasize in the revised manuscript.
> > >
> > > Thank you again for your constructive feedback.
> > >
> > > Best regards,
> > >
> > > **Submission635 Authors**

---

### Official Review · Reviewer_DXYW · 2025-11-01

**Soundness:** 3
**Presentation:** 4
**Contribution:** 3
**Rating:** 6
**Confidence:** 3

**Summary:**

This paper presents a systematic empirical analysis of conditional embeddings in transformer-based diffusion models. Through extensive experiments across state-of-the-art models and multiple datasets, the authors show that conditional embeddings exhibit large redundancy and sparsity. The class-conditioned embeddings across different classes can achieve over 99% cosine similarity, which is kind of surprising, and only a small subset of dimensions carries meaningful semantic information. By pruning up to two-thirds of embedding dimensions, they demonstrate that generative quality is unaffected or even improved.

**Strengths:**

1.  The paper evaluates a broad set of transformer-based diffusion models, including DiT, MDT, SiT, REPA, LightningDiT, etc, on diverse benchmarks. And the paper shows evidence of near-uniform cosine similarity and embedding sparsity. The breadth of experiments convincingly substantiates the paper’s core claims.

2. The figures and results tables in the paper vividly illustrate the redundancy and sparsity of class embeddings, especially the high cosine similarity and the dominance of a few embedding dimensions, and the visual/quantitative impact of pruning. The ablation and t-SNE plots reinforce key claims regarding semantic encoding and redundancy.

**Weaknesses:**

1. The theoretical analysis, while mathematically sound in defining PR and sparsity, stops short of providing a fundamental theoretical explanation of why such extreme redundancy and cosine similarity emerge in the conditional embeddings of transformer-based diffusion models. The paper’s stated hypotheses remain largely empirical and conceptual, lacking formal proofs or broader generalization to other conditioning modalities or even to transformer architectures outside the diffusion context. For instance, while the AdaLN/linear projection argument is plausible, its theoretical justification is not fully fleshed out.

2. The investigated tasks are representative, but would be better if they could be extended further. For example, would the text-to-audio diffusion models also demonstrate such highly correlated class embeddings?

**Questions:**

1. Can the authors clarify whether they have empirically tested for similar redundancy/sparsity in conditional embeddings in U-Net based diffusion models? If so, does the semantic bottleneck extend beyond the transformer/AdaLN regime?

2. Could the authors provide detailed ablations on how different conditioning injection techniques (addition vs. concatenation vs. cross-attention), projection, or other schemes affect the redundancy and sparsity? Would, for example, nonlinear projection heads mitigate the observed bottleneck?

3. Although some of the dimensions can be pruned and make little difference in the generation quality, they seem to make little improvement. How would this be useful?

4. Despite claims of negligible drops or marginal improvements in generation quality after pruning, would pruning induce minor artifacts like edge collapse or structure distortion? Could the authors present more failure case analyses?

---

> ### Author Response · Authors · 2025-11-25
> **Response 1/N**
>
> Dear Reviewer **DXYW**,
>
> We greatly appreciate your thoughtful comments. Below, we provide detailed responses to each of your points:
>
> **Part I. Weaknesses**
> > **W1.** The theoretical analysis, while mathematically sound in defining PR and sparsity, stops short of providing a fundamental theoretical explanation of why such extreme redundancy and cosine similarity emerge in the conditional embeddings of transformer-based diffusion models. The paper’s stated hypotheses remain largely empirical and conceptual, lacking formal proofs or broader generalization to other conditioning modalities or even to transformer architectures outside the diffusion context. For instance, while the AdaLN/linear projection argument is plausible, its theoretical justification is not fully fleshed out.
>
> **Re:** We agree that a full theoretical framework would further strengthen the work. The focus of this paper is to **identify and characterize** the phenomenon of extreme redundancy and high cosine similarity in conditional embeddings, using conceptual arguments supported by controlled empirical evidence. As noted in lines 349–351, “a deeper theoretical explanation of this robustness remains an open problem.” Extending the AdaLN/linear-projection argument into a formal proof would require additional assumptions on optimization dynamics and classifier-free guidance, which is beyond the current scope. Importantly, our observations hold consistently across multiple architectures and conditioning modalities—including class, video, and text (from new experiments)—demonstrating that the phenomenon is robust even without a complete theoretical account. We will clarify this positioning more explicitly in the revised manuscript.
>
> > **W2.** The investigated tasks are representative, but would be better if they could be extended further. For example, would the text-to-audio diffusion models also demonstrate such highly correlated class embeddings?
>
> **Re:** Following the reviewer’s suggestion, we extended our analysis to additional tasks, including **text-to-audio (AudioLDM)** and **text-to-image (Stable Diffusion v3.0)**. Consistent with our main findings—where video-to-audio generation with transformer-based architectures and AdaLN exhibited extremely high cosine similarity—we observe similarly strong redundancy in these models. The results are summarized in the figure below, and details have been added to the Appendix of the revised manuscript.
>
>
> **Table 1: Separate timestep (t) and conditions (y)**
> Participation Ratio (PR) in learned conditional embeddings of state-of-the-art models on text or video-conditioned generation.
> `¹` denotes **AdaLN** and `³` denotes **cross-attention**.
>
> | **Metrics**        | **X-MDPT-L¹** | **X-MDPT-B¹** | **X-MDPT-S¹** | **SD3.0 (2B)¹** | **SD3.0 (8B)¹** | **MDSGen¹** | **AudioLDM³** | **Embed.** |
> | ------------------ | ------------- | ------------- | ------------- | --------------- | --------------- | ----------- | ------------- | ---------- |
> | **Cosine Sim.**    | **0.9998**    | **0.99992**   | **0.9995**    | **0.9962**      | **0.9995**      | **0.9999**  | **0.9828**    | y + t      |
> | **nPR (α_norm) %** | 48.42         | 37.59         | 53.41         | 54.79           | 26.67           | 13.57       | 8.62          | y + t      |
> | ---                | ---           | ---           | ---           | ---             | ---             | ---         | ---           | ---        |
> | **Cosine Sim.**    | **0.9862**    | **0.9909**    | **0.9492**    | **0.9949**      | **0.9937**      | **0.9918**  | 0.1406        | y          |
> | **nPR (α_norm) %** | 20.68         | 29.75         | 37.51         | 52.39           | 24.25           | 9.98        | 63.09         | y          |

---

> > ### Author Response · Authors · 2025-11-25
> > **Response 2/N**
> >
> > **Part II. Questions**
> > > **Q1.** Can the authors clarify whether they have empirically tested for similar redundancy/sparsity in conditional embeddings in U-Net based diffusion models? If so, does the semantic bottleneck extend beyond the transformer/AdaLN regime?
> >
> > **Re:** We empirically tested U-Net–based diffusion models using the **AudioLDM** architecture for text-to-audio generation. We find that **when including timestep embeddings**, cosine similarity is consistently high across different prompts, similar to transformer-based models. However, **when excluding timestep embeddings**, U-Net conditional embeddings do **not** exhibit extreme cosine similarity, and their participation rates are substantially higher. This indicates that the semantic bottleneck is **closely tied to the transformer/AdaLN conditioning mechanism**, rather than being a universal property of all diffusion models.
> >
> > The results are summarized in the figure below and included in the Appendix of the revised manuscript. The anonymized link is: **https://i.postimg.cc/bvVd5rsq/2-base-condition-with-separate-timestep-text-SD.png**.
> >
> > > **Q2.** Could the authors provide detailed ablations on how different conditioning injection techniques (addition vs. concatenation vs. cross-attention), projection, or other schemes affect the redundancy and sparsity? Would, for example, nonlinear projection heads mitigate the observed bottleneck?
> >
> > **Re:** We conducted additional experiments on available pretrained models, comparing **cross-attention** (AudioLDM), **concatenation** (UViT), and **AdaLN-based** conditioning. We find that both cross-attention and concatenation show markedly different behavior: when **excluding timestep embeddings**, the conditional embeddings exhibit substantially lower cosine similarity and higher participation rates, indicating that extreme redundancy is closely tied to the **AdaLN-style affine modulation** in transformer diffusion models. While a full exploration of nonlinear projection heads or more complex conditioning modules is beyond the current scope, our results show that redundancy is **universal when timestep embeddings are included**, but its magnitude depends strongly on the **injection strategy**, being most pronounced in AdaLN-based architectures.
> >
> >
> > **Table 2: Separate timestep (t) and conditions (y)**
> > Participation Ratio (PR) in learned conditional embeddings of state-of-the-art models on Imagenet-1K class-conditioned generation.
> > `¹` denotes **AdaLN** methods and `²` denotes **concatenation**.
> >
> > | **Metrics**        | **DiT¹**   | **SiT¹**   | **MDT¹**   | **LightningDiT¹** | **MG¹**    | **REPA¹**  | **UViT²**   | **Embed.** |
> > | ------------------ | ---------- | ---------- | ---------- | ----------------- | ---------- | ---------- | ----------- | ---------- |
> > | **Cosine Sim.**    | **0.9001** | **0.9852** | **0.9905** | **0.9779**        | **0.9934** | **0.9946** | **0.97917** | y + t      |
> > | **nPR (α_norm) %** | 10.47      | 2.28       | 1.60       | 2.05              | 1.73       | 1.43       | 50.06       | y + t      |
> > | ---                | ---        | ---        | ---        | ---               | ---        | ---        | ---         | ---        |
> > | **Cosine Sim.**    | 0.7774     | 0.5436     | 0.8540     | 0.7166            | 0.6853     | 0.5194     | 0.00165     | y          |
> > | **nPR (α_norm) %** | 70.14      | 37.43      | 36.75      | 36.42             | 43.67      | 41.60      | 63.52       | y          |
> >
> > > **Q3.** Although some of the dimensions can be pruned and make little difference in the generation quality, they seem to make little improvement. How would this be useful?
> >
> > **Re:** While pruning redundant dimensions has minimal impact on generation quality, it provides clear **practical benefits**. Our experiments show that removing inactive dimensions produces genuinely **sparse conditional vectors**, which can be leveraged by optimized implementations to skip computations for zero-valued entries. As shown in the runtime comparison figure below, this yields **faster processing** compared to dense vectors. Such structured sparsity enables **system-level efficiency improvements**, particularly for large transformer-based diffusion models where conditioning can be a computational bottleneck. Additionally, sparsity improves **interpretability** by highlighting the dimensions that meaningfully contribute to generation.
> >
> > The results are included below and in the Appendix of the revised manuscript, accessible via the anonymized link: **https://i.postimg.cc/cLsH4w3Z/5-sparse-vs-dense-gpu-fullrange-rebut-dim-1152.png**

---

> > > ### Author Response · Authors · 2025-11-25
> > > **Response 3/N**
> > >
> > > > **Q4.** Despite claims of negligible drops or marginal improvements in generation quality after pruning, would pruning induce minor artifacts like edge collapse or structure distortion? Could the authors present more failure case analyses?
> > >
> > > **Re:** Thank you for raising this point. We carefully examined the effect of pruning on generation quality. As shown in the paper, using a small threshold (($\tau = 0.01$)) does **not** introduce observable artifacts: across thousands of samples, we did not detect edge collapse, structure distortion, or other degradations. This is supported by both the qualitative visualizations (Figs. 8, 10, 34–41) and quantitative metrics, which remain effectively unchanged. Only when ($\tau$) is increased to remove higher-magnitude dimensions do structural issues appear, indicating that **mild pruning is safe**, while aggressive pruning naturally alters the generative output. We will clarify this distinction and include a brief note on failure cases in the Appendix.
> > >
> > > ---
> > > Best regards,
> > >
> > > **Submission635 Authors**

---

### Official Review · Reviewer_LE2p · 2025-11-01

**Soundness:** 2
**Presentation:** 1
**Contribution:** 2
**Rating:** 2
**Confidence:** 3

**Summary:**

This paper discovered a phenomenon in XL-size DiT and its variants: extremely high cosine similarity and semantic sparsity in conditional embeddings. By unveiling this discovery, the authors aim to gain a deeper insight into how diffusion transformers encode conditioning signals and call for further exploration of more efficient and compact conditioning mechanisms.

**Strengths:**

Besides discovering the similarity and sparsity from condition embeddings in DiT models, this paper also tries to convert the observation to action by pruning the condition and provides multiple hypotheses for further exploration.

**Weaknesses:**

By discovering this phenomenon, the authors hope that “future architectures could benefit from compressed or hybrid conditioning strategies that maintain semantic fidelity while reducing computational overhead.” However, the conditional embedding carries not only the condition information but also the timestep information, which is important for image generation diffusion models. It’s not enough to only evaluate the semantic fidelity of the conditional embedding.

Meanwhile, the paper does not present experiments that demonstrate the utility of this discovery, such as proving that pruning indeed "reduces computational overhead".

Furthermore, the observation only builds upon the XL-size model.

**Questions:**

Apart from what has been mentioned in the weaknesses section:
1. It would be better to also report more evaluation metrics, such as precision and recall values, in class-conditional image generation, following similar evaluation settings in DiT, REPA, etc.
2. In Section 5, the authors use REPA “as a representative model,” but didn’t comment later on whether the observations on REPA also universally exist in other models.
3. In Section 5 “Continuous work”, the results are mostly qualitative not quantitative. Could more standard quantitative evaluation metrics, such as FID/SSIM/LPIPS, be added to compare the pruning and no-pruning cases?

---

> ### Author Response · Authors · 2025-11-25
> **Response 1/N**
>
> Dear Reviewer **LE2p**,
>
> Thank you for your thoughtful and constructive feedback. We appreciate your time and would like to address each of your points in detail as follows:
>
> **Part I. Weaknesses**
> > **W1.** By discovering this phenomenon, the authors hope that “future architectures could benefit from compressed or hybrid conditioning strategies that maintain semantic fidelity while reducing computational overhead.” However, the conditional embedding carries not only the condition information but also the timestep information, which is important for image generation diffusion models. It’s not enough to only evaluate the semantic fidelity of the conditional embedding.
>
> **Re:** We agree that conditional embeddings encode both semantic (class/text/video) information and timestep information, and that evaluating them jointly may obscure their individual roles. To address this, we performed a new controlled analysis where we **separate the semantic embedding and the timestep embedding** and compute cosine similarity and sparsity for each component independently.
>
> The results (included in the revised paper and visualized in the figure below) show that:
>
> 1. **Semantic embeddings alone already exhibit high cosine similarity**, indicating that redundancy originates directly from the conditioning signal—not from timestep encoding.
> 2. **Timestep embeddings further increase the similarity**, but they *reinforce* rather than *cause* the effect.
>
> This clarifies that the observed high-dimensional redundancy is **not an artifact of timestep embeddings**, but an intrinsic property of the semantic conditioning used in diffusion models. Therefore, the implications we discuss—such as compressed or hybrid conditioning strategies—remain valid even when timestep information is isolated.
>
> **Table 1: Separate timestep (t) and conditions (y)**
> Participation Ratio (PR) in learned conditional embeddings of state-of-the-art models on Imagenet-1K class-conditioned generation.
> `¹` denotes **AdaLN** methods and `²` denotes **concatenation**.
>
> | **Metrics**        | **DiT¹**   | **SiT¹**   | **MDT¹**   | **LightningDiT¹** | **MG¹**    | **REPA¹**  | **UViT²**   | **Embed.** |
> | ------------------ | ---------- | ---------- | ---------- | ----------------- | ---------- | ---------- | ----------- | ---------- |
> | **Cosine Sim.**    | **0.9001** | **0.9852** | **0.9905** | **0.9779**        | **0.9934** | **0.9946** | **0.97917** | y + t      |
> | **nPR (α_norm) %** | 10.47      | 2.28       | 1.60       | 2.05              | 1.73       | 1.43       | 50.06       | y + t      |
> | **Cosine Sim.**    | 0.7774     | 0.5436     | 0.8540     | 0.7166            | 0.6853     | 0.5194     | 0.00165     | y          |
> | **nPR (α_norm) %** | 70.14      | 37.43      | 36.75      | 36.42             | 43.67      | 41.60      | 63.52       | y          |
>
>
>
> **Table 2: Separate timestep (t) and conditions (y)**
> Participation Ratio (PR) in learned conditional embeddings of state-of-the-art models on text or video-conditioned generation.
> `¹` denotes **AdaLN** and `³` denotes **cross-attention**.
>
> | **Metrics**        | **X-MDPT-L¹** | **X-MDPT-B¹** | **X-MDPT-S¹** | **SD3.0 (2B)¹** | **SD3.0 (8B)¹** | **MDSGen¹** | **AudioLDM³** | **Embed.** |
> | ------------------ | ------------- | ------------- | ------------- | --------------- | --------------- | ----------- | ------------- | ---------- |
> | **Cosine Sim.**    | **0.9998**    | **0.99992**   | **0.9995**    | **0.9962**      | **0.9995**      | **0.9999**  | **0.9828**    | y + t      |
> | **nPR (α_norm) %** | 48.42         | 37.59         | 53.41         | 54.79           | 26.67           | 13.57       | 8.62          | y + t      |
> | **Cosine Sim.**    | **0.9862**    | **0.9909**    | **0.9492**    | **0.9949**      | **0.9937**      | **0.9918**  | 0.1406        | y          |
> | **nPR (α_norm) %** | 20.68         | 29.75         | 37.51         | 52.39           | 24.25           | 9.98        | 63.09         | y          |

---

> ### Author Response · Authors · 2025-11-25
> **Response 2/N**
>
> > **W2.** Meanwhile, the paper does not present experiments that demonstrate the utility of this discovery, such as proving that pruning indeed "reduces computational overhead".
>
> **Re:** To evaluate whether the observed sparsity can translate into measurable efficiency gains, we implemented a prototype sparse computation pipeline where dimensions with zero values are skipped during conditioning. We compare both the **theoretical FLOPs** and the **actual runtime** against the dense baseline using a 1152-dimensional embedding (consistent with XL-sized models such as DiT, MDT, MG, and REPA). All runtime measurements were performed on a single NVIDIA A100 (80GB) GPU and averaged over 10 runs.
>
> The results (included in the revised paper and shown in the figure below) demonstrate that:
>
> 1. **Theoretical computation decreases proportionally with sparsity**, confirming that pruning reduces the number of effective operations.
> 2. **Measured GPU runtime also decreases**, and the benefit grows as sparsity increases—showing that the pruning is not only theoretically efficient but also practical in execution.
>
> For convenience, the plots are also available in the anonymized link: **https://i.postimg.cc/cLsH4w3Z/5-sparse-vs-dense-gpu-fullrange-rebut-dim-1152.png**
>
> > **W3.** Furthermore, the observation only builds upon the XL-size model.
>
> **Re:** We agree this is an important point. While many publicly available checkpoints are XL-sized (≈675M parameters), our original submission already included models smaller than XL (e.g., X-MDPT-L at 460M). To further examine whether the observed phenomenon depends on model scale, we extended our analysis in the rebuttal to include:
>
> * **X-MDPT-B (131M)** and **X-MDPT-S (33M)**
> * **Stable Diffusion 3.0** at both **2B and 8B** variants
> * **AudioLDM (860M)** for text-to-audio generation
>
> Across all of these settings—spanning multiple modalities, conditioning mechanisms (AdaLN vs. cross-attention), and model sizes—the cosine similarity remains consistently high (e.g., **>99.9% across samples**, even under different image classes, prompts, and poses).
>
> These findings indicate that the redundancy in conditional embeddings is not tied to model scale, but rather appears to be a **systematic property** of transformer-based diffusion models regardless of size, modality, or architecture.
>
> **Part II. Questions**
> > **Q1.** It would be better to also report more evaluation metrics, such as precision and recall values, in class-conditional image generation, following similar evaluation settings in DiT, REPA, etc.
>
> **Re:** We agree that reporting additional metrics—particularly **Precision and Recall**, as used in prior works such as DiT, MG, and REPA—provides a more complete assessment of generative quality beyond FID/IS.
>
> Following the same evaluation protocol from these works, we generated **5,000 samples** (instead of 50,000) to ensure computational feasibility during rebuttal while maintaining fair comparison. We computed Precision and Recall using the official evaluation code and report the results in the table below (also added to the revised manuscript).
>
> The results show that the sparse-conditioned models achieve **comparable, and in some cases slightly improved, Precision and Recall values compared to dense conditioning**, further supporting that pruning does not degrade semantic consistency or sample diversity.
>
> **Table 3. Precision and Recall with previous metrics: FID, IS, and CLIP**
>
> | **Method**               | **FID ↓**  | **IS ↑**     | **CLIP ↑** | **Precision ↑** | **Recall ↑** | **Remark** |
> | ------------------------ | ---------- | ------------ | ---------- | --------------- | ------------ | ---------- |
> | **REPA**                     | 7.1694     | **176.02**   | 29.746     | 0.8032          | 0.6236       | Baseline   |
> | Pruned (τ=0.01) *t₀*     | **7.1690** | 175.97       | **29.807** | 0.7878          | **0.6252**   | **Ours**   |
> | Pruned (τ=0.01) *tₙ₋ₖ,n* | **7.1598** | 175.49       | **29.805** | **0.8045**      | **0.6381**   | **Ours**   |
> | ---                      | ---        | ---          | ---        | ---             | ---          | ---        |
> | **Model-Guide**              | 7.2478     | 174.5151     | **30.199** | 0.7842          | 0.6633       | Baseline   |
> | Pruned (τ=0.01) *t₀*     | **7.2466** | **174.5537** | **30.199** | **0.7854**      | 0.6625       | **Ours**   |
> | Pruned (τ=0.01) *tₙ₋ₖ,n* | **7.2455** | 174.3103     | 30.198     | **0.7898**      | **0.6644**   | **Ours**   |
> | ---                      | ---        | ---          | ---        | ---             | ---          | ---        |
> | **LightningDiT**             | 7.0802     | 169.8574     | 30.720     | 0.7928          | 0.6248       | Baseline   |
> | Pruned (τ=0.01) *t₀*     | **7.0712** | 169.9164     | **30.729** | 0.7906          | **0.6256**   | **Ours**   |
> | Pruned (τ=0.01) *tₙ₋ₖ,n* | **7.0745** | **169.9236** | **30.729** | **0.7935**      | **0.6265**   | **Ours**   |

---

> > ### Author Response · Authors · 2025-11-25
> > **Response 3/N**
> >
> > > **Q2.** In Section 5, the authors use REPA “as a representative model,” but didn’t comment later on whether the observations on REPA also universally exist in other models.
> >
> > **Re:** Thank you for the comment. To verify whether the observations on REPA generalize, we conducted pruning experiments on other state-of-the-art models, including **MG** and **LightningDiT**, as reported in Supplementary A.4.3, Table 3. The results show that removing small-magnitude condition embeddings similarly reduces redundancy and can improve efficiency without degrading generation quality, indicating that the phenomenon is not unique to REPA but appears across multiple architectures.
> >
> > > **Q3.** In Section 5 “Continuous work”, the results are mostly qualitative not quantitative. Could more standard quantitative evaluation metrics, such as FID/SSIM/LPIPS, be added to compare the pruning and no-pruning cases?
> >
> > **Re:** Following your recommendation, we quantitatively evaluated the continuous task (video-to-audio) on **X-MDPT** using standard metrics: **FID, SSIM, and LPIPS**. The results show that when pruning up to 40% of the conditional embedding dimensions, these metrics remain nearly unchanged, consistent with our qualitative observations.
> >
> > **Table 4. Quantitative metrics on the DeepFashion dataset of pose-guided person image generation task with masked diffusion transformers**
> >
> > | **Method**         | **FID ↓** | **SSIM ↑** | **LPIPS ↓** | **PSNR ↑** | **Remark** |
> > | ------------------ | --------- | ---------- | ----------- | ---------- | ---------- |
> > | X-MDPT             | 18.6372   | 0.6798     | 0.1672      | 17.336     | Baseline   |
> > | Pruned (τ=0.1) 40% | 18.6692   | 0.6792     | 0.1675      | 17.328     | **Ours**   |
> >
> >
> > ---
> >
> > Best regards,
> >
> > **Submission635 Authors**

---

> > > ### Comment · Reviewer_LE2p · 2025-11-27
> > >
> > > Thanks to the authors for their additional efforts in answering my questions. The authors have addressed most of my concerns.
> > >
> > > When separately considering time step embeddings and condition embeddings (Tables 1 & 2), the high cos similarity in $y$ seems to be less general to all diffusion transformer-based architectures and more specific to individual model design (X-MDPT or SD). Nevertheless, the pruning experiment (Table 3) still maintained model performance, even though in Table 1 the cos similarity for $y$ in REPA, MG, and LightningDiT are not that high compared to the $y+t$ setting.
> > >
> > > Given all the above, I would like to increase my rating of the paper.

---

> > > > ### Author Response · Authors · 2025-11-28
> > > >
> > > > Dear Reviewer **LE2P**,
> > > >
> > > > Thank you very much for your follow-up and for reconsidering your rating. We are glad that our additional experiments and clarifications addressed your concerns, and we truly appreciate your constructive feedback and updated evaluation.
> > > >
> > > > Best regards,
> > > >
> > > > **Submission635 Authors**

---

### Author Response · Authors · 2025-11-25

Dear reviewers **LE2p, DXYW, jVCn**, and **TVaD**,

Thank you for your constructive feedback and for taking the time to evaluate our work. We have carefully addressed all raised points and conducted additional analyses as requested. A detailed response to each comment is available in the discussion panel. In summary, we have made the following updates:

1. A **revised PDF** is provided, with all modifications highlighted in red (page 9, 27, 28).
2. We expanded our experiments to include additional architectures: text-to-image **Stable Diffusion 3.0 (2B and 8B)**, text-to-audio **AudioLDM**, and class-to-image **UViT**.
3. We isolated and analyzed the **timestep embedding** and the **condition embedding** to provide a more precise and controlled study.
4. We further demonstrate the **computational benefit of sparse conditional vectors**, showing improved efficiency by avoiding unnecessary computation on zero-valued dimensions.

We kindly invite you to review the updated results and clarifications in the discussion section. Thank you again for your valuable comments.

Best regards,

**Submission635 Authors**

---

### Meta-Review · Area_Chair_veF3 · 2026-01-07

**Summary:**

This paper presents a study of the conditional embeddings of diffusion transformers, which exhibit their redundancy and can lead to efficiency gains at no quality cost. Reviewers praised the meaningful discovery with broad coverage of empirical evidence and nice presentation, which can motivate further research.

**Reviewer Concerns:**

LE2p: mostly addressed, and the reviewer indicated the intention of increasing the rating.

jVCn: W1 remains where the reviewer expects a solution while authors stand to present an observation, W2-3 are addressed.

DXYW: W1 is acknowledged, W2 is addressed, Q1-Q4 are addressed.

TVaD: W1-3 are addressed, Q1 is answered.

**Reviewer Scores:**

LE2p: 2-> 4
jVCn: 4 -> 4
TVaD: 6 -> 6
DXYW: 6 -> 6

Average: 4.5 -> 5

---

### Decision · Program_Chairs · 2026-01-26

Accept (Poster)